# Proteome-wide detection of *S*-nitrosylation targets and motifs using bioorthogonal cleavable-linker-based enrichment and switch technique

Ruzanna Mnatsakanyan[1], Stavroula Markoutsa[1], Kim Walbrunn[1], Andreas Roos[1,2], Steven H.L. Verhelst[1,3] & René P. Zahedi[1,4,5]

Cysteine modifications emerge as important players in cellular signaling and homeostasis. Here, we present a chemical proteomics strategy for quantitative analysis of reversibly modified Cysteines using bioorthogonal cleavable-linker and switch technique (Cys-BOOST). Compared to iodoTMT for total Cysteine analysis, Cys-BOOST shows a threefold higher sensitivity and considerably higher specificity and precision. Analyzing S-nitrosylation (SNO) in S-nitrosoglutathione (GSNO)-treated and non-treated HeLa extracts Cys-BOOST identifies 8,304 SNO sites on 3,632 proteins covering a wide dynamic range of the proteome. Consensus motifs of SNO sites with differential GSNO reactivity confirm the relevance of both acid-base catalysis and local hydrophobicity for NO targeting to particular Cysteines. Applying Cys-BOOST to SH-SY5Y cells, we identify 2,151 SNO sites under basal conditions and reveal significantly changed SNO levels as response to early nitrosative stress, involving neuro(axono)genesis, glutamatergic synaptic transmission, protein folding/translation, and DNA replication. Our work suggests SNO as a global regulator of protein function akin to phosphorylation and ubiquitination.

---

[1] Leibniz-Institut für Analytische Wissenschaften-ISAS-e.V., Otto-Hahn-Str. 6b, 44227 Dortmund, Germany. [2] Department of Neuropediatrics, Centre for Neuromuscular Disorders in Children, University Hospital Essen, University of Duisburg-Essen, Hufelandstr. 55, 45122 Essen, Germany. [3] Laboratory of Chemical Biology, Department of Cellular and Molecular Medicine, KU Leuven - University of Leuven, Herestraat 49Box 8023000 Leuven, Belgium. [4] Gerald Bronfman Department of Oncology, Jewish General Hospital, McGill University, 5100 de Maisonneuve Blvd. West, Montreal, Quebec H4A 3T2, Canada. [5] Segal Cancer Proteomics Centre, Lady Davis Institute, Jewish General Hospital, McGill University, 3755 Côte Ste-Catherine Road, Montreal, Quebec H3T 1E2, Canada. Correspondence and requests for materials should be addressed to R.P.Z. (email: rene.zahedi@ladydavis.ca)

Cysteine residues have a key involvement in modulation of the function, subcellular localization, enzymatic activity and interactions of proteins[1,2], both in their unmodified and various post-translationally modified (PTM) forms. The transient and target-selective nature of Cys PTMs allows them to act as redox-switches, which play an essential role in redox signaling and regulation of cellular redox homeostasis[3,4]. Among reversible Cys PTMs S-nitrosylation (SNO) has been shown to function as a redox-switch in several (patho)physiological events such as ischemia–reperfusion, cardioprotection, neurogenesis, and synaptic transmission[5–10]. Over the past decades, the growing number of substrates reported across protein classes is in agreement with the ubiquity of SNO-mediated regulatory events[11]. The functional diversity of SNO in its wide range of targets depends on both its localization, e.g., active site and allosteric Cys, as well as its life span. The durability of S–NO linkage[12] could be influenced by the presence of stabilizing/destabilizing groups (ionizing, aromatic, and hydrophobic species) in the vicinity[13]. Protein conformation and local microenvironment seem to be essential for directing and catalyzing SNO formation. The enzymatic nature of SNO and de-nitrosylation is increasingly appreciated in target selectivity and specificity of SNO-mediated redox signaling[14,15]. A growing number of SNO proteins are recognized as transnitrosylases (NO donors) that transfer their NO group to the free Cys of target proteins (NO acceptors)[16–18]. SNO-free thiol homeostasis is maintained by the protein denitrosylase thioredoxin (Trx) system and specialized low-molecular-weight SNO (i.e., S-nitrosoglutathione (GSNO) and SNO-coenzyme) reductase systems[14]. Although the biological significance of endogenous SNO and other reversible Cys modifications is well-recognized, system-wide studies are hindered by several analytical challenges, most importantly the vast diversity of Cys PTMs, their reversibility and lability (potential loss and reshuffling during sample preparation/analysis), as well as low abundance. In recent decades, numerous mass spectrometry (MS)-based analytical strategies have been development to face these challenges[19–25]. Most widespread approaches for Cys PTM analysis utilize an indirect detection via the biotin switch technique (BST) or derivatives thereof (ST)[19,26]. The initial steps of these assays include blocking of all the free thiols, followed by selective reduction of modified Cys and switch with a stable functional group, which also serves as a handle for enrichment. Examples of broadly applied BST/ST assays include isotope-coded affinity tags (ICAT)[22], resin-assisted capture (RAC)[21], and iodoTMT[20]. However, each of these methods has certain limitations, such as the bulky nature of the biotinylated tag for ICAT, possible reshuffling of reduced Cys during binding to thiopropyl-sepharose resin and on-bead digestion for disulfide-exchange-based RAC, as well as poor specificity of anti-TMT antibody-based enrichment for iodoTMT. These challenges were partially addressed by the development of enrichment methods, utilizing photocleavable and azobenzene-cleavable linkers[24,25,27,28].

In this paper, we present Cys-BOOST, a method for Cys analysis using ST with enrichment by a Dde-biotin-azide bioorthogonal cleavable-linker, and amine-reactive TMT labeling for quantitative LC-MS[2] (MS[3])-based analysis. We demonstrate the performance of Cys-BOOST by enrichment of the Cys peptides from tryptic HeLa digests with high specificity and sensitivity. The direct comparison between Cys-BOOST and iodoTMT shows superior performance of Cys-BOOST in terms of both the number of identified (25,019 vs 9966) and quantified (18,074 vs 7201) Cys peptides, technical reproducibility (RSD of 9% vs 36%) and specificity (98% vs 74%). SNO analysis in GSNO-treated and control HeLa cell extracts reveals thousands of SNO targets and allows the identification of proteome-wide GSNO-mediated nitrosylation consensus motifs. These motifs demonstrate on a large scale the contribution of acid–base catalysis and local hydrophobicity to the thus far elusive target selectivity of SNO. Applying Cys-BOOST for the identification of SNO targets in SH-SY5Y cells with and without treatment with 100 μM of the cell-permeable NO donor S-Nitroso-N-Acetyl-D,L-Penicillamine (SNAP) reveals significant regulation of SNO levels in proteins involved in neurogenesis, axonogenesis, glutamatergic synaptic transmission, cadherin binding, NADH metabolic process, protein folding, translation initiation, and DNA replication.

## Results

**Cys-BOOST workflow.** Here we introduce Cys-BOOST, a highly sensitive and specific method for quantitative LC-MS-based analysis of Cys PTMs (Fig. 1).

**Comparison of Cys-BOOST and iodoTMT.** To compare the efficiency of enrichment and the reproducibility of Cys-BOOST and iodoTMT, the total captured Cys residues from HeLa cell lysate were analyzed by both methods (Fig. 2). Reversibly modified Cys were reduced with Tris(2-carboxyethyl)phosphine hydrochloride (TCEP), the lysate was divided into six equal aliquots, and all free Cys (freshly reduced and initially free) were alkylated by iodoacetamide (IAA)-alkyne ($n = 3$) or iodoTMT ($n = 3$), respectively. Next, proteins were digested with trypsin. For Cys-BOOST peptides were labeled with TMT 10plex™ to enable quantification based on the TMT reporter ion intensities, which is intrinsically possible for iodoTMT. In the Cys-BOOST protocol, the Cys peptides were enriched using the Dde-biotin-azide linker and chemical elution via cleavage of the Dde bond. IodoTMT-labeled peptides were enriched via specific binding to an anti-TMT antibody followed by elution at acidic pH. The complexity of the eluates for both methods was reduced before liquid chromatography–mass spectrometry (LC-MS/MS) by on-tip pH 10 fractionation (Fig. 2a). Using iodoTMT-based enrichment 9966 Cys peptides and 3446 background peptides were identified, corresponding to an enrichment specificity of 74% (Fig. 2b, Supplementary Data 1). In contrast, using Cys-BOOST with the same samples, we identified 25,019 Cys peptides and only 581 background peptides, which corresponds to a specificity of 98% and a concurrent 2.5-fold increase in the number of Cys peptides (Fig. 2c, Supplementary Data 2). In addition, the total Cys peptides enriched by Cys-BOOST showed 69% overlap with iodoTMT peptides (Supplementary Figure 1). Notably, we were able to compare the technical reproducibility of the enrichment for both methods by labeling each of the three technical replicates with different TMT channel and by avoiding the pooling of the replicates throughout the enrichment, thus obtaining RSD of 9% vs 36% for Cys-BOOST and iodoTMT, respectively. Although the number of quantified peptides is 2.5 × higher for Cys-BOOST, the boxplots of the scaled TMT reporter intensities (abundances) of all quantified Cys containing peptides show a much narrower distribution and more reproducible mean values for Cys-BOOST than for iodoTMT (Fig. 2d). In general, Cys-BOOST displays excellent technical reproducibility, which opens the possibility to use it for complex studies involving > 10 samples, by combining multiple TMT sets with a common normalizer-channel present in each set. We compared the average-scaled TMT reporter intensities (sum of all TMT reporter intensities of Cys containing PSMs/number of Cys containing PSMs) of all Cys containing PSMs quantified by either Cys-BOOST or iodoTMT (Fig. 2e). Despite the higher number of Cys peptides for Cys-BOOST, which may come along with the identification of many low abundant peptides, the average-scaled TMT intensity observed for Cys-BOOST was around four times higher. This indicates a considerably higher recovery, which comes along with more

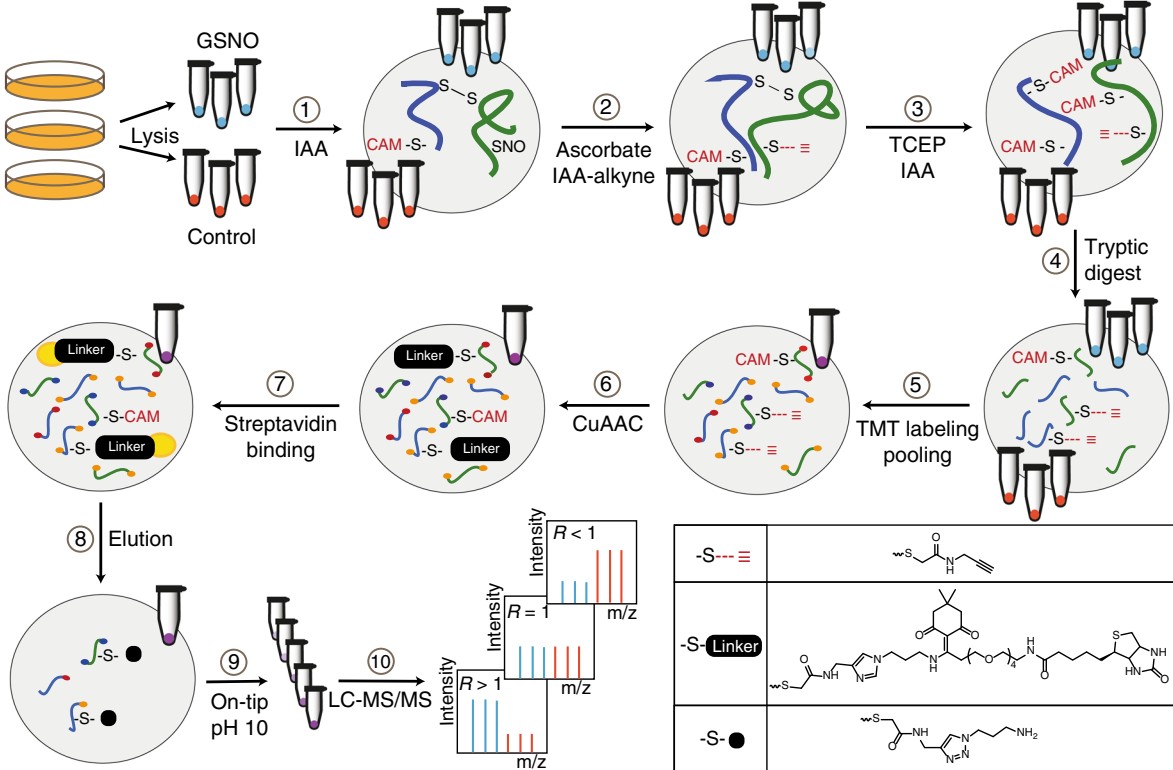

**Fig. 1** Schematic representation of Cys-BOOST. The workflow is exemplified for SNO analysis of untreated ($n = 3$) and GSNO-treated ($n = 3$) HeLa cell extracts. Cys-BOOST incorporates the following steps: (1) irreversible blocking of all free thiols with IAA, (2) switch of specifically reduced Cys with alkyne-derivatized iodoacetamide reagent (IAA-alkyne), (3) reduction and alkylation of the remaining (endogenously) oxidized Cys, (4) protein digestion with trypsin, (5) TMT 10plex™ labeling of Lys residues and N-termini for multiplexed quantitative analysis, (6) conjugation of the Dde-biotin-azide linker to IAA-alkyne labeled peptides via copper(I)-catalyzed alkyne-azide cycloaddition (CuAAC), (7) affinity binding of the Cys peptides to streptavidin beads via the biotin group of the Dde-biotin-azide linker, (8) release of the target peptides with 2% hydrazine by single step mild chemical cleavage of the Dde bond, (9) on-tip pH 10 fractionation of the eluate before (10) LC-MS/MS analysis

precise quantification. Consequently, even with the very stringent criteria applied for data analysis, Cys-BOOST allowed the quantification of 60 Cys peptides per μg of HeLa lysate (compared with 24 for iodoTMT), which, to our knowledge, makes it the most sensitive method for analyzing Cys peptides to date (Fig. 2f). Cys-BOOST proved to be four times more sensitive compared with the recently developed TiO$_2$-based CysPAT, which identified 15 Cys peptides per μg of HeLa lysate[23].

**SNO analysis in GSNO-treated and non-treated HeLa extracts.** After demonstrating that Cys-BOOST allows highly efficient qualitative and quantitative analysis of total Cys peptides, we applied it for the analysis of one of the more challenging Cys PTMs; nitrosylation, which was shown to function as a redox-switch in a number of cardiovascular and neuronal pathophysiological conditions[29]. Although in recent decades multiple analytical strategies were developed for SNO, proteome-wide analysis still represents a significant challenge. We adapted Cys-BOOST for SNO analysis by incorporating the well-established ST using specific reduction of SNO by sodium ascorbate[19,26,30]. To exclude the presence of free Cys prior to ascorbate reduction, leading to false-positive identification of SNO sites, we verified the completeness of free Cys blocking in the initial step of our ST conditions (Supplementary Figure 2–4, Supplementary Data 3–4).

SNO analysis was performed for untreated ($n = 3$) and in vitro GSNO-stimulated ($n = 3$) HeLa cell extracts. SNO peptides were reduced by sodium ascorbate and switched with IAA-alkyne. The following steps were performed according to the Cys-BOOST

workflow (Fig. 1). After TMT labeling, the untreated and GSNO-treated samples were pooled, and the enrichment was performed for a single multiplexed sample. The eluate was then fractionated by on-tip RP at pH 10, and analyzed by LC-MS/MS.

The high sensitivity of Cys-BOOST allowed us to discover proteome-wide SNO targets and consensus motifs. We identified 3632 SNO proteins with 9314 SNO peptides and 8304 unique SNO sites (Supplementary Data 5), out of which 6247 SNO peptides were confidently quantified (false discovery rate (FDR) ≤ 1%, ptmRS SNO site localization probabilities ≥ 99%, reporter ion co-Isolation threshold ≤ 20, average reporter S/N ≥ 10). Mapping the identified SNO proteins to the HeLa proteome based on normalized spectral abundance factors (NSAF)[31], showed that the SNO proteome covers the whole dynamic range of the HeLa proteome (Supplementary Figure 5a). Next, we sought to identify consensus motifs for Cys sites having differential reactivity to GSNO. The motifs were identified for three distinct groups defined by their $R$ [GSNO]:[Control] ratios. SNO sites showing downregulation and no significant changes after GSNO treatment ($R ≤ 1.5$) were considered as GSNO non-reactive and they correspond to endogenous SNO. The decreased SNO levels might be explained by reduction or exchange of endogenous SNO for instance by free glutathione, after its NO was transferred to another Cys. We considered SNO sites with $1.5 < R < 6$ ($p$ value ≤ 0.05) as GSNO mild-reactive and SNO sites with $R ≥ 6$ ($p$ value ≤ 0.05) as GSNO hyper-reactive ($p$ values defined by analysis of variance; ANOVA). From 480 mappable GSNO non-reactive sites 362 (75%) were matched by motif-x[32] to one of the motifs in Fig. 3a (Supplementary Table 1a). The high

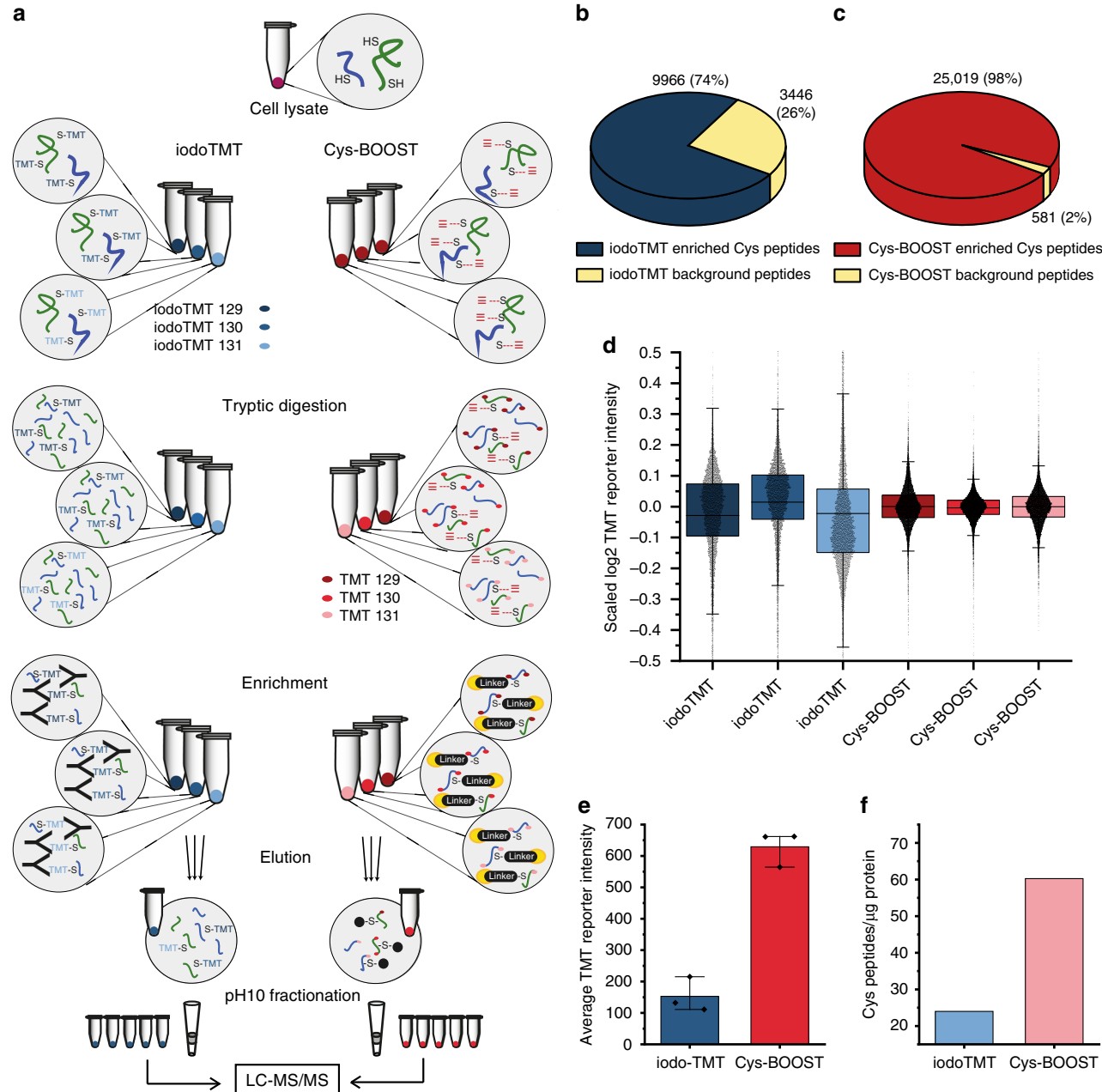

**Fig. 2** Total Cys analysis by iodoTMT and Cys-BOOST. **a** Comparison of iodoTMT and Cys-BOOST workflows for total Cys peptides analysis from 100 μg of HeLa lysate per replicate. **b** and **c** pie charts showing the share of the target (Cys containing) and background peptides. **d** Boxplots of scaled log$_2$ TMT reporter intensities of all quantified Cys containing peptides display a substantially more accurate and precise quantification for Cys-BOOST as compared with iodoTMT (center line: mean; box limits: upper and lower quartiles; whiskers: correspond to the highest or lowest respective value or if the lowest or highest value is an outlier it is 1.5 × interquartile range (IQR)). **e** Average TMT reporter intensities of Cys-containing PSMs (error bars show the minimum and maximum of $n = 3$ replicates) indicate a higher recovery by Cys-BOOST. **f** Number of Cys containing peptides quantified per μg of HeLa lysate. Source data are provided as a Source Data file

percentage of matching SNO sites demonstrates that the motifs are highly representative for the group. These motifs are dominated by the presence of hydrophobic amino acids leucine (Leu; L), isoleucine (Ile; I), valine (Val; V), and aromatic phenylalanine (Phe; F), particularly the position −4 (four amino acids N-terminal of the Cys) proves to be significant. From 1894 mappable GSNO mild-reactive sites, 283 (15%) were matched to two motifs, both containing lysine (Lys; K) either at the position +6 or −6 from the SNO site (Fig. 3b, Supplementary Table 1b). From 3446 mappable GSNO hyper-reactive sites, 2826 (82%) were matched to one of the motifs presented in Fig. 3c

(Supplementary Table 1c). Again, these motifs are highly representative for the GSNO hyper-reactive group. Interestingly, 19 of 20 motifs defined for hyper-GSNO reactive SNO sites contain at least one of the two acidic amino acids, aspartate (Asp; D) and glutamate (Glu; E). Moreover, at the positions ±3 and ±4 both Asp or Glu containing versions of the motifs were individually found by motif-x (Fig. 3c).

A STRING network analysis of the proteins possessing either significantly downregulated (Fig. 4a) or the 500 strongest upregulated (Fig. 4b) SNO sites illustrates the association of SNO to stress response, ubiquitin mediated proteolysis,

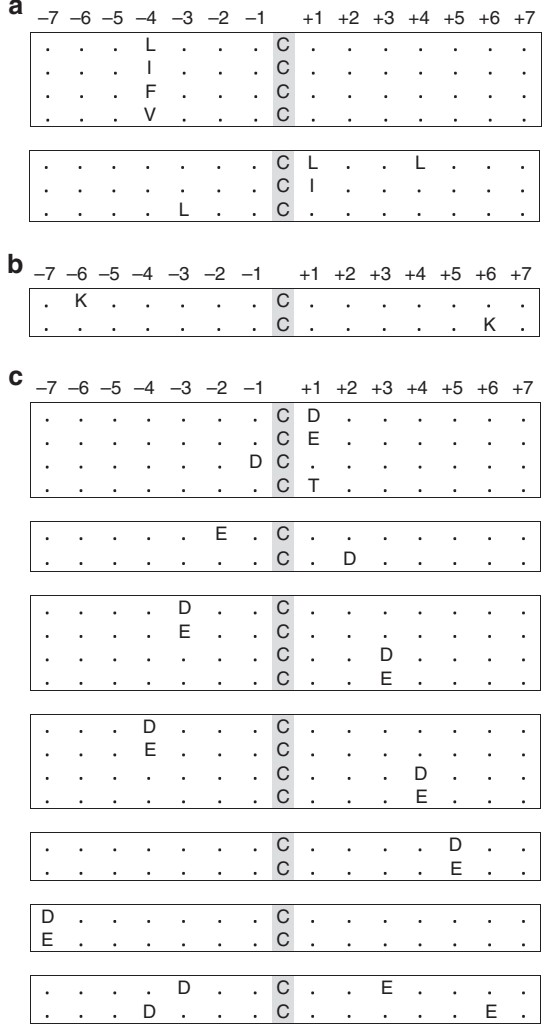

**Fig. 3** SNO consensus motifs identified by motif-x v1.2. Significance $< 1 \times 10^{-6}$, fold increase $\geq 1.59$. **a** Motifs for GSNO non-reactive ($R \leq 1.5$), **b** GSNO mild-reactive ($1.5 < R < 6$; ANOVA $p$ value $\leq 0.05$) and **c** GSNO hyper-reactive ($R \geq 6$; ANOVA $p$ value $\leq 0.05$) SNO sites. $P$ values derived from $n = 3$ biological replicates. Source data are provided as a Source Data file

translation, splicing, DNA replication, GTPase-mediated signaling and vesicle-mediated transport pathways.

Heat shock proteins HSP90-alpha, HSP90-beta, HSPA8, HSPA4, HSPA4L, HSPA14, HSPH1, mitochondrial HSPD1, and HSPA9 were identified with multiple SNO sites. Remarkably, different (even adjacent) Cys sites of the same protein had highly varying levels of SNO, as depicted for of HSP90-alpha (Fig. 5a), which is known for its nitric oxide synthase (NOS) regulatory activity[33]. Here, the SNO level of Cys 420 was fivefold decreased, whereas the SNO level of Cys 374 was 14-fold increased. We furthermore detected Cys 597 and Cys 589 from the C-terminal domains of HSP90-alpha and HSP90-beta, respectively, as endogenously nitrosylated, whereas the adjacent Cys were detected as free in agreement with former studies[33,34]. Besides, four novel SNO sites on HSP90-alpha and three novel sites on HSP90-beta were identified. The SNO level of Cys 310, which is located in a structurally similar region of HSPA4, HSPA4L, and HSPH1 was significantly decreased in all three proteins (Fig. 5a).

SNO seems to make a considerable contribution to the regulation of the ubiquitin–proteasome system. In total, 44 E3 ubiquitin–protein ligases, as well as multiple proteasome subunits were identified as SNO targets. SNO sites on the Cys proteases: caspase 1 (Cys 136), 2 (Cys 343), 3 (Cys 170), and 4 (Cys 109), were found in the vicinity of the active sites, but active site Cys themselves were not identified as nitrosylated, pointing to SNO-mediated allosteric regulation. Only caspase 8 was detected with multiple SNO sites (Cys 164, 203, 360, 426) including the active site Cys 360. Allosteric regulation could also explain the SNO sites on 86 phosphatases and phosphatase subunits detected by Cys-BOOST.

Nucleic acid binding is one of the most-prominent molecular functions matched by STRING for the SNO proteins, indicating the important role of SNO in transcription and translation. Indeed, 43 RNA and four DNA helicases were found to be nitrosylated. Taking into the account that the human genome encodes for only 64 non-redundant RNA helicases there is a substantial enrichment. SNO levels of Cys 198 of spliceosome RNA helicase DDX39B (HFILDEC$_{198}$DKMLEQ) and Cys 197 of ATP-dependent RNA helicase DDX39A (HFVLDEC$_{197}$DKM-LEQ) located at structurally similar regions, inside the conserved DECD motif, were found significantly decreased in both proteins ($R = 0.2$ and $R = 0.3$, respectively).

Interestingly, STRING networks of proteins with both significantly downregulated and upregulated SNO levels show an enrichment of coatomer protein complex subunits and other proteins involved in vesicle mediated transport (more pronounced for upregulated sites), which is in agreement with formerly detected regulation of exocytosis by SNO[35,36]. From proteins involved in DNA replication, minichromosome maintenance (MCM) proteins appear to be more susceptible to SNO. Three of the four Cys present in the MCM domains of MCM2, MCM3, MCM5, and all four of MCM7 were detected as nitrosylated, with varying $R_{\text{[GSNO]:[Control]}}$.

Cys-BOOST detected multiple SNO proteins involved in redox regulation including thioredoxin, peroxiredoxin, and glutaredoxin family proteins. Cys 62 and 69 of thioredoxin were identified as SNO, which is in agreement with X-ray crystallography analysis of the GSNO-treated protein[37]. Nitrosylated Cys 343 and 312 of another redox sensitive protein, protein disulfide isomerase (PDIA1), are located in a noticeable distance from active sites Cys 53 and 56, which were not detected as nitrosylated themselves, again inferring a potential allosteric regulation via SNO. Moreover, both Cys 343 and 312 are located in immediate structural proximity but display extremely varying SNO levels (Fig. 5a).

Although we did not expose our cells to hypoxia, we were able to detect SNO sites on several important hypoxia players. Two novel SNO sites were detected on hypoxia upregulated protein 1 (HYOU1). Cys 352 and 805 are surface exposed and are found in the small disordered regions between two alpha-helixes (Fig. 5a). Both sites show no significant change after GSNO treatment and thus represent possible endogenous SNO targets. We also identified SNO sites on hypoxia-responsive transcriptional activators: Cys 404 of CREB-binding protein (CBP), which is located at the CH1 domain, Cys1621 and 1247 of p300, from which Cys1621 is located at the histone acetyltransferase domain. Another interesting target of nitrosylation represents Cys 169 of Cbp/p300-interacting transactivator 4 (CITED4), which has a C-terminal localization similar to Cys 800 of hypoxia inducible factor 1-alpha (HIF1α), a known SNO target (Fig. 5b)[6]. We identified all nine Cys of NOS-interacting protein, five of which were identified as nitrosylated and the other four as free Cys (Fig. 5c).

**SNO analysis in SNAP-treated and control SH-SY5Y cells.** Next, we applied Cys-BOOST for the quantitative analysis of SNO targets in untreated and SNAP-treated SH-SY5Y cells. The

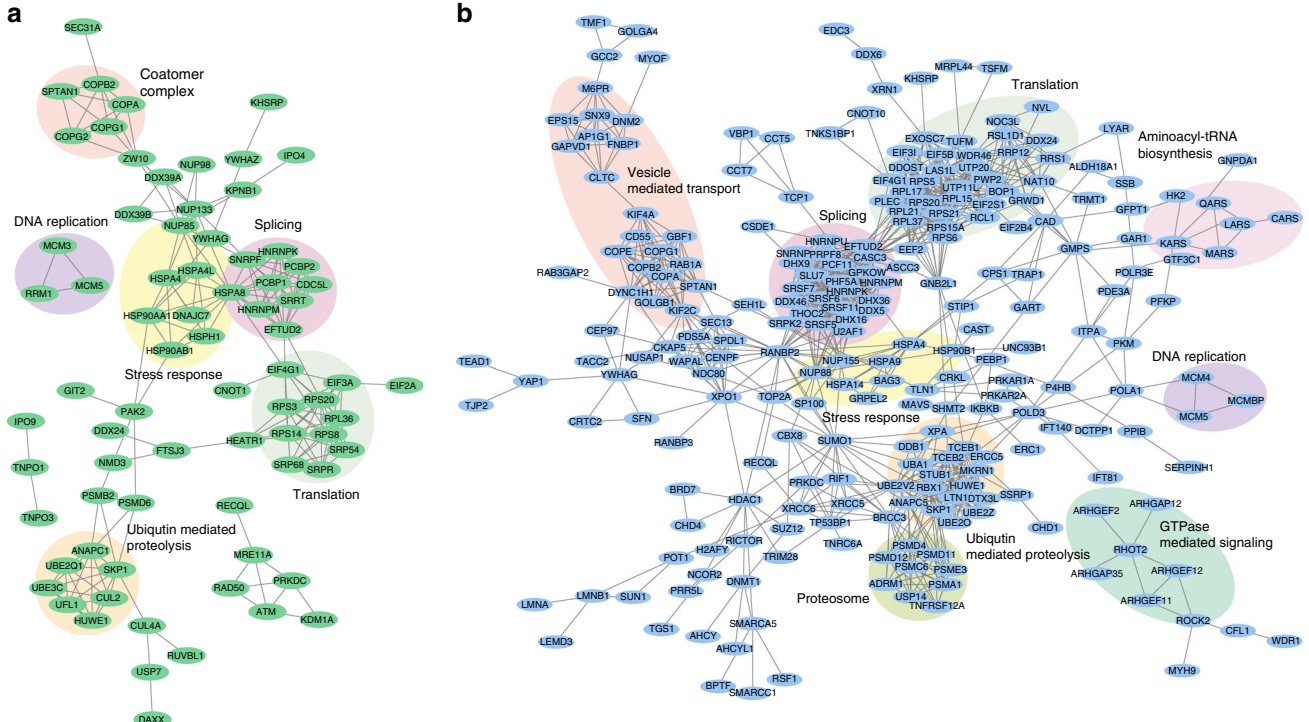

**Fig. 4** The STRING networks of proteins with regulated SNO sites after GSNO treatment of HeLa extracts. $N = 3$ biological replicates. **a** Proteins with significantly downregulated SNO sites, **b** the 500 proteins with the strongest upregulated SNO sites. Both networks indicate the role of SNO in splicing, translation, ubiquitin mediated proteolysis, DNA replication, cellular response to stress, vesicle-mediated transport, GTPase-mediated signaling and aminoacyl-tRNA biosynthesis pathways. Source data are provided as a Source Data file

neuroblastoma SH-SY5Y cell line is widely used as neuronal in vitro system[38], for instance in Parkinson's disease and Alzheimer's disease research. To study the effects of short-term nitrosative stress, SH-SY5Y cells were treated for 5 min with 100 μM SNAP ($n = 4$) or incubated with phosphate-buffered saline (PBS; $n = 4$) as controls (i.e., basal conditions). We identified 2439 SNO peptides (1855 confidently quantified), representing 2158 unique SNO sites on 1443 proteins. In total, 2151 SNO sites were identified in both, control (basal) and SNAP-treated samples, covering four orders of magnitude of the dynamic rage of the SH-SY5Y proteome (Supplementary Data 6, Supplementary Figure 5b). A total of 65 and 39 SNO sites were significantly upregulated ($\geq 1.5$-fold, ANOVA $p$ value $\leq 0.05$) and downregulated ($\geq 1.5$-fold, ANOVA $p$ value $\leq 0.05$), respectively (Fig. 6b), with an enrichment of proteins involved in protein folding, translation, DNA replication, NADH metabolic processes, and cadherin binding (Fig. 6c). Proteins involved in NADH metabolic processes and DNA replication showed predominantly downregulated SNO. Notably, several proteins involved in neuro(axono)genesis (DOCK7, MAP1B, PSIP1), axon guidance (DPYSL4), and synaptic transmission (GLUL) were detected with significantly altered (mostly upregulated) SNO sites (Supplementary Data 6), suggesting that our approach allowed deep insights into the SNO-mediated dynamic regulation of proteins with neuroprotective properties upon the presence of pathophysiological-relevant stress conditions.

## Discussion
Cys-BOOST incorporates a switch technique using IAA-alkyne for exchanging the reversibly modified thiols with a stable functional group, which consecutively serves as a handle for introduction of the bioorthogonal Dde-biotin-azide linker at the peptide level. This allows efficient enrichment by using streptavidin and specific elution via chemical cleavage of the Dde bond. In combination with TMT labeling Cys-BOOST enables the highly effective LC-MS-based quantitative analysis of Cys PTMs. The here-presented strategy resolves issues, such as less-efficient ionization and increased complexity of MS/MS spectra related to the bulky nature of common thiol labeling reagents such as ICAT. Using Cys-BOOST, the samples can be processed via traditional bottom–up proteomics workflows, involving in-solution or filter-aided sample preparation[39,40], thus avoiding on-bead digestion procedures common for many Cys PTM analytical strategies[41,42]. One of the key advantages of the bead-free digestion is the possibility to introduce the TMT labels at an earlier step of the workflow, allowing pooling of the samples before the enrichment. Thus, the CuAAC can be conducted on the peptide level, followed by streptavidin enrichment of a single, pooled sample, which results in improved reproducibility and robustness of the analysis. Notably, iodoTMT reagents allow the introduction of the TMT tag even at an earlier stage, during the switch of reversibly modified thiol. Nevertheless, the subsequent anti-TMT antibody-based enrichment suffers from lower specificity (74%) and sensitivity (24 Cys peptides quantified per μg HeLa lysate) when compared with Cys-BOOST. In contrast, the strong biotin/streptavidin affinity in Cys-BOOST allows for stringent washes, consequently efficient depletion of background peptides. The specificity of the enrichment relies furthermore on the highly specific chemical cleavage of the Dde bond of the linker for elution, which yields a single cleavage product, fully compatible with LC-MS/MS and LC-synchronous precursor selection (SPS) analysis. Moreover, the linker fragment from the cleaved biotin tag in Cys-BOOST yields a diagnostic fragment ion ($m/z$ 230.17), which could be used to enhance the confidence of HCD spectra assignments in accordance with Yang et al.[25]. Collectively, the above-mentioned factors ensure the high specificity (98%), reproducibility (RSD of 9%), and sensitivity (60 Cys

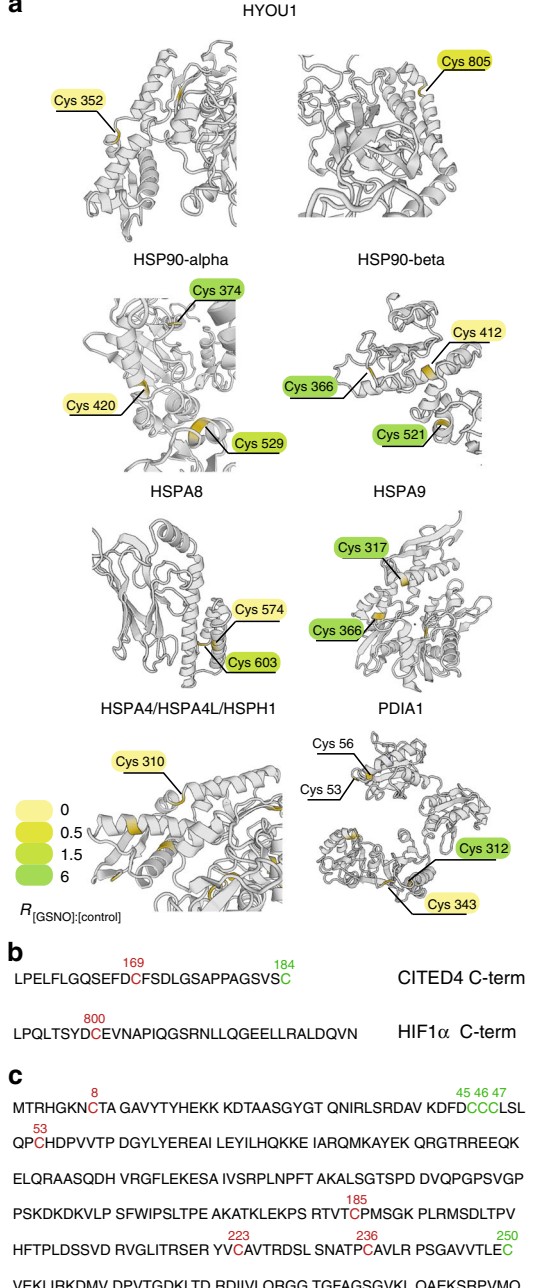

**Fig. 5** SNO proteins relevant in heat shock response and hypoxia. **a** Structures of HYOU1, heat shock proteins (HSP), and PDIA1 with highlighted SNO sites. The color gradient of the highlighted background of Cys symbolizes the $R_{[GSNO]:[Control]}$ ratio. The yellow highlights on the protein structure correspond to the position of the Cys. Protein structures are obtained from SWISS-MODEL[71]. **b** SNO sites (red) of CITED4 (novel site detected by Cys-BOOST) and HIF1α (detected formerly) both located at C-termini and putatively involved in activation of hypoxia-responsive genes. Cys 184 of CITED4 was detected as free. **c** All nine Cys of NOSIP were identified, four of which were free Cys (green) and five were nitrosylated (red). Source data are provided as a Source Data file

peptides quantified per μg of HeLa lysate) of Cys-BOOST. In addition, the superior performance of Cys-BOOST compared to iodoTMT reinforces Cys-BOOST as an effective analytical strategy for the analysis of reversibly modified Cys.

Considering the vast biological significance of SNO[3,30,43] and its well-established specific reduction by ascorbate[19,26], we chose

the analysis of SNO as a prominent example to demonstrate the potential of Cys-BOOST for the proteome-wide quantitative analysis of reversible Cys modifications with high sensitivity and high precision. Applying Cys-BOOST for the comparison of in vitro GSNO-treated and non-treated HeLa lysates, enabled us to identify an unmatched number of SNO-modified proteins (3632), peptides (9314), and sites (8304), compared with previous studies (Supplementary Table 2). GSNO participates in protein SNO formation mainly through thiol-to-thiol $NO^+$ transfer. For the bacterial transcription factor OxyR, GSNO-mediated nitrosylation was shown to be facilitated by a sequence motif including flanking acidic/basic and hydrophobic residues[13]. Our quantitative data allowed us to classify the SNO sites according to their reactivity to GSNO and to define sequence motifs for differentially reactive sites. Thus, we classified SNO sites as GSNO hyper-reactive, GSNO mild-reactive and GSNO non-reactive, based on the TMT reporter ion intensity ratios between treated and untreated samples ($R_{GSNO/Control}$). The identification of distinct sequence motifs for these classes demonstrates the appropriateness of the initial grouping. Nevertheless, when discussing SNO motifs the transient nature of SNO should be taken into the account. Indeed, SNO formed via in vitro treatment with GSNO might be exchanged with free thiols and other Cys PTMs. In addition, SNO proteins may act as transnitrosylases and nitrosylate Cys targets that are not susceptible to nitrosylation by GSNO.

The presence of the flanking acidic amino acids in the majority of motifs defined for GSNO hyper-reactive sites might be explained by indirect acid–base catalysis, based on the favorable positioning of the NO group via hydrogen bonding between γ-glutamyl amine of GSNO and the γ-carboxylate of Glu or β-carboxylate of Asp, as illustrated for Cys 199 of OxyR[13]. The occurrence of the Asp at the position + 3 from the active Cys 199 allows positioning of the NO group of GSNO within ~ 4 Å of the Cys 199 free thiol, thus assisting the $NO^+$ transfer[13]. In our data, the prominent motifs with Asp or Glu at ± 3 and ± 4 positions from the hyper-reactive SNO sites suggest proteome-wide occurrence of this principle. Occurrence of Asp or Glu in three or four amino acids distance from SNO sites is in agreement with former studies of SNO motifs based on smaller subset of proteins[18,44,45]. For the motifs with Asp or Glu at positions ± 1 and ± 2 the influence might be explained by direct acid–base catalysis of nitrosylation/de-nitrosylation[3,46–48]. Under physiological pH the deprotonated carboxyl group of Glu or Asp within ~ 6 Å (~ 2 amino acids) from the free Cys might act as a base, withdrawing the proton of the thiol and facilitating nucleophilic thiolate ($RS^-$) formation. Alternatively, the presence of the carboxyl group of Glu or Asp in the intermediate vicinity of the nitrosylation target free Cys might facilitate the protonation of GSNO and consequently promote the donation of its NO group, as demonstrated for methionine adenosyltransferase by Perez-Mato et al.[46]. Next, we identified two motifs for GSNO mild-reactive SNO sites, both possessing Lys at position ± 6 from the SNO site. These Lys motifs were recently identified also by Smith et al.[49] in an SNO analysis of nuclear extracts from rat cortical neurons. The presence of the basic amino acid Lys within ~ 6 Å will enhance the nucleophilicity of the thiol group, thus making the Cys more susceptible to nitrosylation and oxidation in general. However, this mechanism would be possible only within a framework of certain structural conformations that position the Lys closer to the Cys, whereas the theoretical linear distance (calculated using 3.5 Å as amino-acid average length) of the ± 6 Lys would be ~ 20 Å. The combined knowledge of SNO motifs containing characteristic acidic amino acid or basic Lys allows us to discuss the model of nitrosylation via indirect acid–base catalysis not only for GSNO, but also for transnitrosylases. Here, the

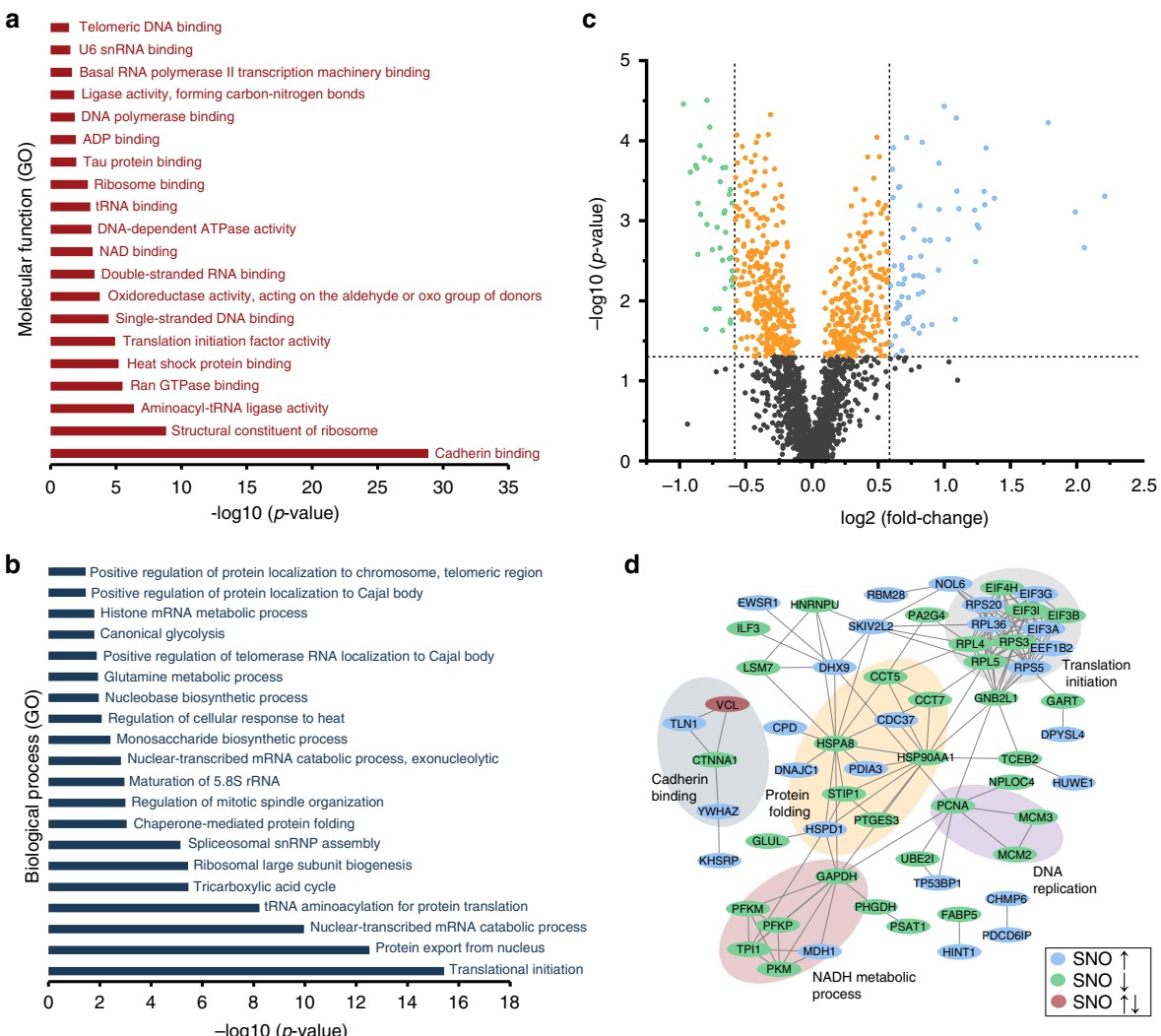

**Fig. 6** SNO analysis of SNAP-treated and non-treated (control) SH-SY5Y cells. **a–b** Top 20-enriched Gene Ontology (GO) terms (PANTHER Overrepresentation Test, http://www.pantherdb.org/, p values defined by Fisher's exact test) of the SH-SY5Y SNO proteome: **a** molecular function and **b** biological process. **c** Volcano plot of SNO peptides in SNAP-treated vs non-treated samples. Green dots ($n = 39$) represent significantly downregulated ($\geq 1.5$-fold, ANOVA p value $\leq 0.05$) and blue dots ($n = 65$) significantly upregulated ($\geq 1.5$-fold, ANOVA p value $\leq 0.05$) SNO peptides. **d** High-confidence STRING network of proteins with significantly changed ($\geq 1.5$-fold, ANOVA p value $\leq 0.05$) SNO levels. Proteins with upregulated SNO sites are marked blue, with downregulated green and with both up- and downregulated red. P values derived from $n = 4$ biological replicates. Source data are provided as a Source Data file

indirect acid–base catalysis could be accomplished by favorable positioning of the SNO group of the NO$^+$ donor protein and the free thiol of the acceptor protein. By assistance of hydrogen bonding between the side chain ε-amino group of ± 6 Lys and the side chain carboxylate of the ±3, ±4, +5, or −7 acidic amino acid, the SNO could be placed in an appropriate distance (~ 4 Å) for transnitrosylation. It should be noticed that the proteins having one of these motifs could act both as NO donor and acceptor, in agreement with the dual function of transnitrosylases. SNO sites with the ± 6 Lys motif might be putative targets of transnitrosylation rather than being directly nitrosylated by GSNO, which could explain their presence in the GSNO mild-reactive group. Apparently, the mechanism of the acid–base assisted nitrosylation/transnitrosylation/de-nitrosylation is dependent on the proximity and structural arrangement of the flanking acidic and basic amino acids to target thiol. On the one hand the acidic/basic residues located in direct proximity (~ 6 Å) of the Cys might increase its nucleophilicty or facilitate the NO release from GSNO or transnitrosylases; on the other hand the acidic/basic residues

located ~ 7–20 Å away from the Cys might catalyze the SNO formation indirectly, via favorable positioning of the NO group of either GSNO or transnitrosylases.

The GSNO non-reactive sites identified in our study should represent the more stable naturally occurring SNO. As evidenced by the consensus motifs of this group, the SNO is shielded by flanking hydrophobic amino acids. The enrichment of hydrophobic amino acids in the motifs of GSNO non-reactive sites might indicate SNO formation by NO directly or by auto-oxidation products of NO (e.g. NO$_2$, N$_2$O$_3$). Several arguments support the relevance of the protein local hydrophobicity for endogenous SNO and SNO formed by NO directly[3,13,29,50]. Primarily, the restricted access of hydrophilic reductants and low molecular weight thiols in hydrophobic milieu inhibits the de-nitrosylation[13]. Next, the hydrolysis of NO auto-oxidation products is inhibited and the intermediate radical species (e.g., NO$^•$, RSNO$^{•-}$) are stabilized in the presence of hydrophobic residues[13,51]. The statement is also supported by endogenous SNO sites found in the juxtamembrane region[52,53]. The decrease

of SNO levels of certain Cys in this group might be a consequence of NO transfer to free glutathione or other free thiols.

The 200 μM in vitro GSNO treatment results in widespread SNO formation. In total, 57% of the proteins were detected with multiple SNO sites. Nevertheless, the detected SNO proteome covers the whole dynamic range of the HeLa proteome emphasizing the selectivity of SNO targets and the high sensitivity of Cys-BOOST. Interestingly, the nitrosylation levels of multiple sites detected on individual proteins diverge significantly, once more reinforcing the high specificity of SNO targets. The occurrence of SNO on a range of protein classes is in agreement with numerous regulatory functions attributed to SNO[29]. Although all 8304 unique SNO sites detected in our study are unlikely to occur in endogenous conditions simultaneously, each one of them represents a potential in vivo SNO target which may occur individually under given circumstances.

To study the biochemical impact of SNO under neuropathologically relevant conditions, we applied Cys-BOOST for the quantitative analysis of in vivo SNO targets in SNAP-stimulated and control neuron-like SH-SY5Y cells. Our rationale was that (i) in neurons physiological levels of SNO notoriously regulate neurogenesis, neural development, synaptic plasticity, and neurotransmitter release[10] and (ii) increased SNO has been linked to activation of neuroprotective properties of proteins[54], whereas (iii) excessive (pathophysiological) levels of SNO, known as nitrosative stress, have been related to neuronal damage and the etiology of neurodegenerative disorders[55,56]. In this context, our data show that short-term nitrosative stress resulted in increased SNO of proteins involved in neurogenesis and axonogenesis. We therefore identified candidates of neuroprotection modulated by SNO, such as DOCK7. It has been shown that DOCK7 is involved in neurogenesis and that the presence of genetic mutations directly impacts the manifestation of neurological diseases[57]. Our data moreover indicate that proteins, such as GOSR1, with a known neuroprotective role against redox-based cytoxicity might be controlled by SNO, rather than only by protein abundance[58]. Other examples are endoplasmic reticulum (ER)-stress and UPR-related proteins such as CDC37, HSPD1, and PDIA3. For PDI, it has been shown that SNO inhibits its ability to prevent neurotoxicity associated with ER-stress and protein misfolding[59] and we could identify two novel SNO sites (Cys 85 and 92) in PDIA3. Hence, our results indicate that SNO of proteins involved in ER-stress and UPR might be a synergistic and synchronous pathobiochemical process in the etiology of neurological diseases as already demonstrated for Alzheimer's disease[60]. The existence of a considerable number of downregulated SNO underlines the importance of activation of protein de-nitrosylation pathways and might indicate activation of these proteins in response to cellular stress via liberation of active site or allosteric Cys[61]. We found reduced SNO in proteins with key functions in metabolic processes such as GAPDH (Cys 247), GLUL (Cys 99), PFKM (Cys 114), PFKP (Cys 123), PKM (Cys 49), and TPI1 (Cys 164). In contrast, MDH1, which produces NADH as a byproduct of its enzymatic activity, was detected with upregulated SNO at Cys 154. These findings once more point to the involvement of SNO in the rerouting of glucose to the pentose phosphate pathway, which generates NADH to support cellular antioxidant defense[62,63]. Moreover, our data indicate a SNO-related control of VCL and CTNNA1 which are both members of the cadherin–catenin complex that is crucial for dendrite and synapse morphogenesis[64].

Collectively, our results demonstrate that Cys-BOOST is an efficient analytical strategy for the quantitative analysis of total Cys and exogenous as well as endogenous SNO, with a promising potential of application to study diverse Cys modifications using reducing agents or alkyne reagents specific for the PTM. As exemplified in SH-SY5Y cells, Cys-BOOST allows the proteome-wide identification of SNO even from low abundance proteins and under basal conditions.

## Methods

**Cell culture and lysis.** HeLa S3 cells (ACC 161, DSMZ, Germany) were cultured in 100 mm tissue culture (TC) dishes to ~ 80% confluency in Dulbecco's Modified Eagle Medium (DMEM) medium supplemented with 10% fetal bovine serum (FBS), 1% penicillin/streptomycin at 37 °C, 5% $CO_2$. Medium was removed and cells were washed with PBS. Cells were harvested with 0.05% Trypsin, 0.02% ethylenediaminetetraacetic acid (EDTA) in PBS, incubated at 37 °C for 5 min. Trypsin was diluted with excess of fresh medium, cells were pelleted for 5 min at $300 \times g$, and the pellets were washed twice with PBS. For total Cys analysis the cell pellet from 100 mm² TC dish was lysed in 500 μL of 1% (w/v) sodium dodecyl sulfate (SDS), 150 mM NaCl, 100 mM 4-(2-hydroxyethyl)-1-piperazineethanesulfonic acid (HEPES) (pH 7.5) supplemented with protease inhibitor cocktail Complete Mini (Roche, Switzerland). For Cys nitrosylation analysis cell pellets from three 100 mm² TC dishes were lysed in 500 μL of HENS buffer (1% (w/v) SDS, 1 mM EDTA, 100 mM neocuproin, 100 mM HEPES (pH 7.5)), supplemented with protease inhibitor cocktail Complete Mini. The lysates were incubated with 7 μL of benzonase nuclease (25 units/μL) (Merck, Germany), and 2 mM $MgCl_2$ for 30 min at 37 °C, followed by centrifugation at $18,000 \times g$ for 30 min at 4 °C. Supernatants were collected and protein concentrations were determined by bicinchoninic acid assay (BCA) (Thermo Scientific). SH-SY5Y cells (11th passage, ACC 209, DSMZ, Germany) were cultured in $n = 8$ 100 mm² TC dishes to 60–80% confluency in DMEM medium supplemented with 10% fetal FBS, 1% penicillin/streptomycin at 37 °C, 5% $CO_2$. Medium was removed and cells were washed with PBS. Fresh DMEM medium with 1% penicillin/streptomycin was added and the cells were starved for 45 min at 37 °C, 5% $CO_2$, with the sterilization-UV light of the incubator switched off. Medium was removed and the cells were washed with PBS. 3 mL of 100 μM SNAP in PBS was added to four TC dishes, whereas 3 mL of PBS was added to the other four TC dishes. Cells were incubated for 5 min at 37 °C, 5% $CO_2$ with the sterilization-UV light of the incubator switched off. The medium was removed, and cells were washed with PBS. Next, cells were lysed on TC dishes with 1 mL of lysis buffer containing 100 mM IAA in HENS supplemented with protease inhibitor cocktail Complete Mini.

**Total Cys analysis for comparison of Cys-BOOST and iodoTMT.** Proteins were precipitated by addition of five volumes of ice-cold acetone followed by incubation for 20 min at −80 °C and centrifugation at $18,000 \times g$ for 30 min at 4 °C; subsequently, the supernatant was carefully removed. The protein pellet was resuspended in 1% SDS to a final concentration of 2 mg/mL. Cys were reduced with 5 mM TCEP for 1 h, at 56 °C, with agitation at 300 rpm. The lysate was divided in $n = 3$ replicates for Cys-BOOST and $n = 3$ replicates for iodoTMT-based enrichment, 100 μg of total protein each (based on BCA assay). The pH was adjusted with 200 mM HEPES, pH 7.5. Cys were alkylated with 12 mM iodoacetamide alkyne (IAA-alkyne) (Setareh Biotech) or 12 mM iodoTMT (Thermo Scientific) for Cys-BOOST and iodoTMT-based enrichment, respectively, for 1 h, at 25 °C, with shaking at 300 rpm in the dark. Proteins were digested using the filter-aided sample preparation (FASP)[40] protocol with modifications. Briefly, samples were diluted fivefold with freshly prepared 8 M urea, 100 mM Tris-HCl at pH 8.5, and transferred to 30 kDa Molecular Weight Cutoff centrifugal filters (Pall). The liquid was passed through the filter at $13,800 \times g$, 25 °C and the same centrifugation conditions were applied for all the following FASP steps. The concentrates contained in the filter devices were washed 3 × with 200 μL of freshly prepared 8 M urea, 100 mM Tris-HCl at pH 8.5 and 3 × with 200 μL of 50 mM triethylamonium bicarbonate (TEAB), pH 8.5. The concentrate was digested in 100 μL of 50 mM TEAB pH 8.5, 0.2 M guanidine hydrochloride (GuHCl), 1 mM $CaCl_2$ and trypsin (Sequencing grade, Promega) in a ratio 20:1 (protein:enzyme) at 37 °C overnight. The digests were collected by centrifugation, and the filters were rinsed with 100 μL of 50 mM TEAB and centrifuged. The digests were acidified with 1% trifluoroacetic acid (TFA) and dried under vacuum. The quality of the digest was ensured using a monolithic column high-performance liquid chromatography-setup[65].

For Cys-BOOST peptides were redissolved in 100 μL of 100 mM TEAB, pH 8.5 and labeled with TMT 10plex™ (Thermo Scientific) according to the manufacturer's instructions. Samples were dried under vacuum and resuspended in 200 μL of 0.1% TFA. Samples were desalted with self-packed stage tips using C18 3 M Empore SPE Extraction Disks (Sigma-Aldrich). In brief, the 200 μL pipette tip was packed with 2 mm of C18 3 M Empore SPE Extraction material and 500 μg of Oligo R3 reversed-Phase Resin (Applied Biosystems) in acetonitrile (ACN). ACN was removed by centrifugation at $500 \times g$, 25 °C. The same centrifugation conditions were applied for all the following steps. The cartridge was equilibrated with 200 μL of 0.1% TFA. The samples were loaded, washed with 100 μL of 0.1% TFA and eluted with 100 μL of 70% ACN, 0.1% TFA. The samples were dried under vacuum. The $n = 3$ replicates labeled with three different TMT channels were kept separate. Dde (1-(4,4-dimethyl-2,6-dioxocyclohex-1-ylidene)ethyl) biotin-azide linker (Jena Bioscience, Germany) was conjugated to IAA-alkyne labeled peptides via CuAAC[66]. The samples were dissolved in 100 μL of 100 mM potassium phosphate buffer, pH 7. CuAAC was performed via stepwise addition of 150 μM Dde-biotin-

azide linker, a premixed solution of 250 μM CuSO₄ (≥ 99%, Sigma-Aldrich) and 1.25 mM Tri(3 hydroxypropyltriazolylmethyl)amine (THPTA) (Sigma-Aldrich), 5 mM sodium ascorbate (≥ 99%, Roth) and 5 mM aminoguanidine hydrochloride. The reaction was incubated for 1 h, at 25 °C. Directly after the CuAAC the samples were incubated with 250 μL of prewashed High Capacity Streptavidin—Agarose Resin (Thermo Scientific) for 1 h, at room temperature (RT), head over tail rotation. The beads were pelleted by centrifugation for 30 s at 500 × g, rested on ice for 5 min and the supernatant was removed. The same procedure was applied for removal of the supernatant in all the following steps of the enrichment. The beads were washed 4 × with 1.5 mL of 2 M urea in PBS, 2 × with 1.5 mL of PBS and 2 × with 1.5 mL of 100 mM Na₂PHO₄, pH 7.5. After each washing step the supernatant was removed. The peptides were released by incubation with 125 μL of 2% hydrazine (98%, Sigma-Aldrich) in 100 mM Na₂PHO₄, pH 7.5 for 1 h, at RT, head over tail rotation. Subsequently, the beads were rinsed with 125 μL of 50 mM ammonium formate (≥ 99%, Sigma-Aldrich) and this wash was combined with the eluate.

For iodoTMT-based enrichment peptide pellets were dissolved in 100 μL of 100 mM Tris-HCl, pH 7.5. The n = 3 replicates labeled with three different iodoTMT channels were kept separate. Each replicate was incubated with 200 μL prewashed Immobilized Anti-TMT™ Antibody Resin (catalogue number 90076, Thermo Scientific Pierce) for 2 h, at RT, head over tail rotation. The beads were pelleted by centrifugation for 30 sec at 500 × g, rested on ice for 5 min and the supernatant was removed. The same procedure was applied for the removal of the supernatant in all the following steps. The beads were washed 3 × with 1.5 mL of 1 M urea in 100 mM Tris-HCl, pH 7.5, 2 × with 1.5 mL of 100 mM Tris-HCl at pH 7.5 and with 1 × 1.5 mL of water. After each washing step the supernatant was removed. The peptides were eluted with 100 μL of 50% ACN, 1% TFA. Eluates were dried under vacuum and resuspended in 100 μL of 20 mM ammonium formate, pH 10.0.

**SNO analysis in GSNO-treated HeLa lysates.** The concentration of the lysates was adjusted to 2 mg/mL with lysis buffer. n = 3 replicates, 300 μg of total protein each, were incubated with 200 μM GSNO (Sigma-Aldrich) in HENS, another n = 3 replicates were incubated with the corresponding volume of HENS for 20 min, at 25 °C, with agitation at 300 rpm. Proteins were precipitated with five volumes of ice-cold acetone as described above. When necessary, the pellet was resolubilized by ultrasonication (Vibra-cell, 15 s, amplitude 40). Samples were resuspended in 150 μL of HENS and incubated with 100 mM IAA for 30 min, at 25 °C with agitation at 300 rpm in the darkness. Excess of IAA was removed by double acetone precipitation, first with five volumes of ice-cold acetone. Afterwards the pellet was dissolved in 150 μL of HENS and precipitated with 10 volumes of ice-cold acetone. Next, samples were prepared for the SNO switch. Pellets were resolubilized in 300 μL of HENS. The SNO switch was performed by incubating the samples with 20 mM sodium ascorbate and 3 mM IAA-alkyne for 1 h, at 25 °C with agitation at 300 rpm in the darkness. Until this point, the samples were protected from light or processed under a lamp only emitting wavelengths > 500 nm. Excess of the reagents was removed by double acetone precipitation first with five volumes, afterwards with 10 volumes of ice-cold acetone. The pellets were dissolved in 100 μL of 1% SDS. The following steps of reduction with 5 mM TCEP, adjustment of pH with 200 mM HEPES pH 7.5, alkylation with 12 mM IAA, tryptic digestion, TMT 10plex™ labeling and desalting were conducted as described above for total Cys analysis by Cys-BOOST. The samples (n = 3 GSNO treated, n = 3 controls) labeled with six channels of the TMT 10plex™ were pooled. Enrichment was performed as described above for total Cys analysis by Cys-BOOST with all steps scaled up three times.

**SNO analysis in SNAP-treated SH-SY5Y cells.** The lysates were ultrasonicated, free thiols were blocked and excess of IAA was removed by double acetone precipitation, as described above. Pellets were resolubilized in 250 μL of HENS. Protein concentrations were determined by BCA. Protein concentration was adjusted to 1 mg/mL with HENS and the SNO switch was performed as above. Excess of the reagents was removed by double acetone precipitation. Afterward, protein pellets were dissolved in 1% SDS to a final concentration of 2 mg/mL and incubated with 5 mM TCEP for 1 h, at 56 °C, with agitation at 300 rpm. The pH was adjusted to 7.5 with 200 mM HEPES. Cys were alkylated with 20 mM IAA for 1 h, at 25 °C, with agitation at 300 rpm in the dark. Excess of the reagents was removed by acetone precipitation with 10 volumes of ice-cold acetone. The protein pellets were resuspended in 125 μL of 50 mM TEAB pH 8.5, 1 mM CaCl₂, 0.2 M GuHCl and trypsin was added (20:1; protein:enzyme). Proteins were digested at 37 °C overnight with agitation at 300 rpm. The digests were labeled with TMT 10plex™ according to the manufacturer's instructions. Samples were dried under vacuum, resuspended in 200 μL of 0.1% TFA, pooled, and desalted. Enrichment was performed as described above for total Cys analysis by Cys-BOOST with all steps scaled up threefold. The eluate was acidified to 1% TFA and desalted with self-packed C18 stage tips as described above. The sample was dried under vacuum and resolubilized in 0.1% TFA for nano-LC-MS³ analysis in synchronous precursor selection (SPS) mode on an Orbitrap Fusion Lumos.

**Global proteome analysis for SH-SY5Y cells.** Five percent (max. 40 μg total protein) of the Cys-BOOST flowthrough was dried under vacuum, resolubilized in

10 mM ammonium formate, fractionated by high pH reversed-phase chromatography at pH 8.0, using an UltiMate 3000 HPLC (Thermo Scientific). Fractionation was performed on a Zorbax 300SB-C18 column, 0.5 × 150 mm, 5 μm particle size column (Agilent) using a binary buffer system; buffer A: 10 mM ammonium formate, pH 8.0 and buffer B: 84% ACN in 10 mM ammonium formate pH 8.0. Peptides were loaded onto the column in buffer A at a flow rate of 12.5 μL/min and separated using the following gradient: 3–15% B in 10 min, 15–55% B in 55 min, 55–95% B in 5 min, 95% B hold for 5 min. In total, 16 fractions were collected at 60 s intervals from min 5 to 75 in a concatenate mode. The fractions were dried under vacuum and resolubilized in 0.1% TFA for nano-LC-SPS analysis.

**On-Tip pH 10 fractionation after the enrichment.** On-tip pH 10 fractionation was performed using self-packed (as described above) desalting cartridges. The cartridges were equilibrated with 200 μL of 20 mM ammonium formate, pH 10.0. The SNO eluate was directly fractionated. The corresponding replicates of Cys-BOOST and iodoTMT eluates, respectively, were pooled before fractionation. The samples were loaded and washed with 200 μL of 20 mM ammonium formate at pH 10.0. Stepwise fractionation was performed using 50 μL of 16%, 20%, 24%, 28%, and 80% ACN in 20 mM ammonium formate, pH 10.0. After each elution step the column was washed with 20 μL of 20 mM ammonium formate, pH 10.0 and this wash was combined with the preceding eluate. Fractions were dried under vacuum and resolubilized in 15 μL of 0.1% TFA for nano-LC−MS/MS analysis.

**LC-MS/MS and LC-SPS parameters.** Peptides were LC separated using Ultimate 3000 RSLCnano systems (Thermo Fisher Scientific). All samples were pre-concentrated using 0.1% TFA on a trap column (100 μm × 2 cm, C18 Acclaim Pepmap viper; Thermo Fisher Scientific) for 5 min at a flow rate of 20 μL/min, followed by separation on the main column (75 μm × 50 cm, C18 Acclaim Pepmap viper; Thermo Fisher Scientific) at 250 nL/min and 60 °C. Binary buffers (A, 0.1% FA; B, 84% ACN, 0.1% FA) and linear gradients of 3–30% B for 180 min (total Cys analysis), 3–35% B for 90 min (HeLa-SNO and SH-SY5Y global proteome analysis), and 3–35% B for 120 min (SH-SY5Y-SNO analysis) were applied for peptide separation. The LC system was online-connected to an Orbitrap Fusion Lumos (total Cys analysis, SH-SY5Y analysis) or a Q Exactive HF (HeLa-SNO analysis) mass spectrometer (both Thermo Scientific), equipped with nano electrospray sources. On the Orbitrap Fusion Lumos MS survey scans were acquired from 300 to 1550 m/z at a resolution of 120,000, using an AGC target value of 2 × 10⁵ and a maximum injection time (IT) of 50 ms. MS/MS scans were acquired in data dependent acquisition (DDA) mode in the Orbitrap at a resolution of 60,000. Precursor ions were selected using an isolation window (IW) of 0.8 m/z, an AGC target value of 5 × 10⁴, an IT of 200 ms with TopS (3 s) option and a dynamic exclusion (DE) of 20 s. Isolated precursors were fragmented using higher energy collision induced dissociation (HCD) with a normalized collision energy (CE) of 40. SPS scans were acquired according to O'Connell et al.[67] with slight modifications. MS survey scans were acquired in the Orbitrap from 375 to 1550 m/z at a resolution of 120,000, using an AGC target value of 2 × 10⁵ and an IT of 50 ms. The top speed option with a cycle time of 3 s was used to select precursors with an IW of 0.4 Da, followed by CID fragmentation. MS² scans were acquired in the ion trap in Turbo mode with an AGC target value of 2 × 10⁴, an IT of 105 ms, a dynamic exclusion of 20 s and normalized CE of 35. From each MS² the top 10 fragment ions (SPS ions) were simultaneously selected with IWs of 0.4 Da for HCD with a CE of 65. MS³ scans were acquired in the Orbitrap at a resolution of 50,000, with an AGC target value of 1.5 × 10⁵ and an IT of 130 ms. On the Q Exactive HF MS, survey scans were acquired from 300 to 1750 m/z at a resolution of 60,000, using an AGC target value of 1 × 10⁶ and an IT of 120 ms. MS/MS scans were acquired at a resolution of 60, 000. For DDA (Top15) precursors were selected with an IW of 0.4 m/z, minimum AGC target value of 2 × 10⁵ an IT of 200 ms and a DE of 30 s. Fragmentation was done using HCD with a CE of 33.

**Data analysis.** MS raw files were processed with Proteome Discoverer (PD) v2.2 (Thermo Scientific), using Sequest HT with a human Uniprot database, (downloaded November 2016, 20,072 target entries) and a common contaminants database. The search for MS² files was performed using mass tolerances of 10 ppm and 0.02 Da for precursor and product ions, respectively. The search for SPS files was performed using mass tolerances of 10 ppm and 0.5 Da for precursor and fragment ions, respectively. MS³ spectra with 20 ppm integration tolerance were used for TMT reporter ion quantification. A maximum of two missed cleavages were allowed for trypsin. For iodoTMT analysis, iodoTMT6plex (+ 329.227 Da) and oxidation of methionine (Mox, +15.9949 Da) were set as dynamic modifications. For Cys-BOOST cl-DDE (cleaved form of the linker) on Cys (+ 195.112 Da) and Mox were set as dynamic modifications, TMT 10plex (+ 229.163 Da) on N termini and lysines were set as static modifications. For Cys-BOOST based SNO analysis, carbamidomethylation of Cys (CAM; + 57.021 Da) was set as an additional dynamic modification. For SH-SY5Y global proteome database search TMT 10plex, Mox, and CAM were set as modifications. Percolator was used for FDR estimation and data was filtered at ≤1% FDR on PSM and peptide levels. Site probabilities of Cys dynamic modifications were determined using ptmRS[68]. Only unique Cys peptides that passed the FDR criteria and had site localization probabilities for iodoTMT or cl-DDE ≥ 99% were considered. Quantification was done

based on the TMT reporter ion intensities, only spectra with average reporter S/N threshold ≥ 10 were considered for quantification, for $MS^2$ analysis an additional co-Isolation threshold ≤ 20 was applied. For the total Cys analysis normalization of the quantification data to total peptide amount was done using PD v2.2 and the data were scaled by dividing for each peptide the TMT reporter intensities of each channel by the corresponding average TMT reporter intensity of the three replicates. SH-SY5Y SNO and global proteome data was likewise normalized to total peptide amount using PD v2.2. In contrast, the HeLa-SNO data was processed without normalization, owing to the significant fold changes present between GSNO-stimulated and untreated samples in majority of the quantified SNO peptides. In addition, low (< 5%) RSD of reporter ion intensities were observed for background peptides present in the Cys-BOOST flow through (1 μg measured by LC-MS/MS), showing that the protein amounts in all samples were equivalent (Supplementary Data 5). For the HeLa-SNO data, the low abundance resampling mode of the PD v2.2 was applied, which compensates for the missing values of the control TMT channels, by replacing them with random values sampled from the lower 5% of the detected values. Peptides carrying CAM on Cys and Mox were not considered for quantification. P values for all experiments were defined by one-way ANOVA (individual proteins) hypothesis test using PD v2.2.

**SNO motif analysis.** The SNO motif analysis was performed using motif-x v1.2[32,69]. For all quantified Cys peptides the $R_{[GSNO]:[Control]}$ ratio was calculated and peptides were divided into three groups. We considered SNO sites with $R \leq 1.5$, i.e., downregulated and non-significantly changed sites as GSNO non-reactive sites. Next, we considered SNO sites with $1.5 < R < 6$ (p value ≤ 0.05) as GSNO mild-reactive and $R \geq 6$ (p value ≤ 0.05) as GSNO hyper-reactive sites. P values were defined by ANOVA. Each group was processed separately. The peptides were centered at the SNO residue, motif width was set to 15 amino acids, significance to $< 1 \times 10^{-6}$, foreground format MS/MS and the human proteome was used as a reference database. All identified motifs were considered.

**Protein network analysis.** Protein interaction networks were generated using STRING v10.5. For HeLa-SNO analysis considering proteins with (i) significantly downregulated ($R < 0.5$, p value ≤ 0.05) SNO sites and (ii) the 500 unique proteins having the strongest upregulated SNO sites ($R \geq 14.6$, p value ≤ 0.05); using the highest confidence setting. For SH-SY5Y-SNO analysis: considering proteins with significantly changed (≥ 1.5-fold, p value ≤ 0.05) SNO levels using high confidence setting. P values were defined by ANOVA. The networks were visualized using Cytoscape (http://www.cytoscape.org/).

**Reporting summary.** Further information on research design is available in the Nature Research Reporting Summary linked to this article.

## Data availability
The mass spectrometry proteomics data have been deposited to the ProteomeXchange Consortium via the PRIDE[70] partner repository with the dataset identifiers PXD011131 and PXD012485. The source data underlying Fig. 2b–f, 3a–c, 4a, b, 5a–c, 6a–c, and Supplementary Figs. 1, 2, 3a, b, 4, and 5a, b are provided as a Source Data file. A reporting summary for this Article is available as a Supplementary Information file. All other data supporting the findings of the study are available from the corresponding author on reasonable request.

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

## Acknowledgements

This study was supported by the Ministerium für Kultur und Wissenschaft des Landes Nordrhein-Westfalen, the Regierende Bürgermeister von Berlin - inkl. Wissenschaft und Forschung, and the Bundesministerium für Bildung und Forschung. R.P.Z. is also grateful to the German Research Foundation DFG (ZA 639/4-1) and to Genome Canada and Genome BC GTP platform funding for operations and technology development: 264PRO.

## Author contributions

R.M. designed the experiments, with the insights from R.P.Z., S.M. and S.H.L.V. R.M. and K.W. performed analytical work. R.M. performed data analysis. R.P.Z. coordinated the research. R.M. and R.P.Z. wrote the manuscript with editing and review from S.M., S.H.L.V. and A.R.

## Additional information

**Competing interests:** The authors declare no competing interests.

