## [Peer Review File · Nature Communications]

Reviewers' comments:

Reviewer #1 (Remarks to the Author):

The authors report a workflow called Cys-BOOST, which is applied to the analysis of S-nitrosylated cysteine residues (SNO) in proteins. There is a substantial literature indicating the significance of SNO as a possible physiologic redox regulator that connects to the chemistry of nitric oxide and related reactive nitrogen species. A major obstacle to understanding the biology of SNO modifications is a reliable, sensitive and specific analytical method to measure SNO. This work addresses this challenge and advances in this area would be of broad interest.

The Cys-BOOST method is constructed from several of the key tools that have facilitated recent advances in the analysis of cysteine redox modifications, including Click chemistry probes, cleavable biotin linkers and latest generation mass spectrometry (MS) instruments. An important advantage is that the Click chemistry/capture is done at the peptide level, thereby ensuring an enriched population of Cys peptides and site localization by MS/MS. Despite these improvements, the method is still a biotin-switch approach, which has the advantages and disadvantages of this family of methodologies. The key step conferring specificity for SNO is the reductive removal of SNO with ascorbate, with rapid capture of the liberated thiol with an alkynyl iodoacetamide analog. A limitation of this approach is the possible interference by S-sulfenyl (SOH) sites, which could be reduced and labeled along with genuine SNO. Despite this concern, the analytical work appears to be competently done and the analytical platform presents no major concerns.

The first part of the work is a demonstration of the performance of the Cys-BOOST platform (minus the SNO capture) as a method to analyze cysteinyl peptides. The authors demonstrate that the Click capture methodology outperforms a thiol-labeling and immunoprecipitation method (Iodo-TMT). Good, but not really newsworthy.

The major limitation of this work is that the authors simply treat a HeLa cell lysate with S-nitrosylglutathione (GSNO) and analyze the SNO peptides and sites. They generated a large dataset of putative SNO protein/peptide targets and sites and undertook motif and network analyses to interpret the possible significance of SNO sites. This all seems premature and of perhaps uncertain significance because the labeling experiment itself is of uncertain significance. By mixing HeLa lysate with 200 μ M GSNO, they get a lot of labeled sites, but it's not clear which of these might actually be SNO sites in an intact cell in a defined physiologic state. Indeed, the challenge of mapping and quantifying SNO in situ (i.e., in intact cells) is a more relevant, significant and certainly difficult challenge. It seems possible that the Cys-BOOST platform might be adapted to the in situ SNO

mapping problem, but the authors haven't attempted that yet. What the authors present here is a limited technical demonstration—a milestone on the way to a possibly significant achievement.

Minor point: Putative SNO peptides are apparently those identified with the linker fragment from the cleaved biotin tag. Although the mass shift would be distinctive, such fragments often yield diagnostic product ions in MS/MS (see ref. 20). Do the labeled peptides in the Cys-BOOST workflow yield such characteristic, tag-derived fragment ions? If so, can these be used to enhance the confidence of the assignments?

Daniel C Liebler

Reviewer #2 (Remarks to the Author):

Title: Proteome-wide detection of cysteine (trans-)nitrosylation targets and motifs using bioorthogonal cleavable-linker-based enrichment and switch technique (Cys-BOOST)

Manuscript Summary

In their manuscript, Mnatsakanya et al. describe a modified switch technique for the detection of protein S-nitrosothiols (SNOs). Thousands of S-nitrosylated proteins are identified. Inasmuch as cells express ~10,000 proteins, the data imply that the bulk of the proteome may be subject to SNO control. This point may ultimately merit publication, but the data can be strengthened through additional experimentation.

Major Points

The strategy follows the well-established switch principle for the detection of SNO-proteins. However, rather than directly labeling with thiol-reactive biotin, the authors “switch” with IAA-alkyne, which ultimately reacts with a hydrazine-cleavable biotin linker. The method allows for off-bead protein digestion prior to labeling. The authors also validate differential GSNO reactivity motifs, namely acid-base and hydrophobic motifs. Importantly, the authors claim increased sensitivity

compared with other MS-based techniques for SNO-protein identification. In support of this view, they identify and collate by the far the largest set of S-nitrosylated proteins to date. However, three experiments are needed to support the claims for enhanced sensitivity and physiological relevance: 1. SNO-Cys enrichment and identification should be compared with another technique (e.g. biotin switch or SNO-RAC); 2. an Ascorbate (–) control should be provided; and 3. the authors should use the method to detect endogenous SNO-proteins (e.g. SNO generated in RAW cells stimulated with LPS/IFN or HEK cells transfected with nNOS or eNOS.) Knockdown of GSNO reductase (or tissues from GSNOR mice) would identify SNO-proteins formed by GSNO.

Specifically:

1. The authors should directly compare the Cys-BOOST method with iodoTMT or biotin switch (or SNO-RAC) methods for SNO-Cys identification in order to demonstrate direct superiority of this method.

2. In standard SNO switch methods, -ascorbate is used as a control for blocking efficiency. However the authors do not control for blocking efficiency in SNO-Cys enrichment, but only perform MS on +ascorbate samples, raising the concern that many of the proteins and sites they identify are due to incomplete block of free Cys. The -ascorbate control also serves to control for proteins/peptides that non-specifically bind to streptavidin beads. The authors should repeat SNO-Cys MS experiments comparing +/- ascorbate in order to demonstrate that their reported hit sensitivity is not due to incomplete blocking of free Cys thiols and non-specific bead binding.

3. Exogenous GSNO is representative of nitrosative stress. A focus on endogenously regulated SNO-proteins would provide a more physiological picture; knockdown of GSNO reductase would identify endogenous GSNO-derived SNO-proteins. The authors should compare SNO-Cys from a system in which endogenous SNO levels would be altered (e.g. WT and GSNOR-KO tissue stimulated with LPS/IFN) and, if necessary, knock down GSNOR to assess the role of endogenous GSNO.

4. The authors provide a strategy for assessing potential bias of mass-spec identification, by comparing their nitrosylated protein dataset to a HeLa cell proteomic database with quantified levels of protein abundance. However, they compare only the top 500 proteins in each list to determine that their technique identifies proteins over a 5-order of magnitude difference in expression level. The authors should also compare the complete datasets in order to derive a complete assessment of just how wide a range of proteins is identified by their technique.

5. The authors frame the issue of protein S-nitrosylation as the result of largely non-enzymatic processes, in both the introduction and discussion, ignoring a large and growing literature on enzymes that convert NO to SNO, those that transfer SNO between proteins, and those that remove SNO from proteins and small molecules. Indeed, confusingly, the authors mention some of this literature in the discussion, but frame it in an unusual way. The authors need to reassess the current literature of SNO formation, and update their introduction and discussion accordingly.

6. The authors assume the GSNO mediates S-nitrosylation by transnitrosylation. But they provide no data to support this. Also GSNO will not cross the cell directly. Remove (trans) from title.

Reviewer #3 (Remarks to the Author):

This manuscript describes a method to enrich peptides containing free cysteines. The tool is aimed at deciphering redox chemistry which is a very important for regulation.

The authors are modifying the free cysteines with an alkyne to which, later in the process, a base-labile biotin unit is attached which allows for 'catch-and-release'.

The report is divided into two parts where the first compares this new strategy to a commercial product (IodoTMT) and in the 2nd part the authors are applying the strategy to a biological question. Both experiments are well executed.

Benchmarking: The authors used a fully reduced HeLa lysate to test enrichment. To address technical variability the authors used triplicate. The presented method yielded 25,000 cysteine containing peptides with an enrichment of 98% vs the iodo-TMT strategy that resulted in "only" 10,000 peptides with a much lower enrichment (70%). The authors are naming sensitivity as a reason for the better performance. A Venn diagram in Fig. 2 for the compared approaches will be useful.

Though the antibody based iodo-TMT also is an approach to measure free cysteines many antibody based techniques requires larger starting amounts and are generally less sensitivity. ICAT has been around for many years and I find that a better benchmarking of the proposed strategy would be to compare to such an experiments. Using literature – if available – would be sufficient.

My only concern re. the experiment is how efficient the iodo-alkyne reagent is removed. Such compounds are ultra-reactive and if not removed completely they can result in unwanted alkylation of reduced di-sulfur bridges upon addition of TCEP leading to unwanted side effects and false positives. Can TCEP and IAA (IAA would be in excess of iodo-alkyne) be added simultaneously to minimize the above? I do appreciate that two 'removal steps' are included but including some considerations would be appropriate when discussing the strategy.

Maybe just a display of my ignorance but how does GSNO work? This is an important part of this experiment. If space is a concern, the large section describing the biological findings, using STRING (two full figures) can seem 'light' compared to some of the Nature Comm. reports I have looked at.

Reviewer #1:

The authors report a workflow called Cys-BOOST, which is applied to the analysis of S-nitrosylated cysteine residues (SNO) in proteins. There is a substantial literature indicating the significance of SNO as a possible physiologic redox regulator that connects to the chemistry of nitric oxide and related reactive nitrogen species. A major obstacle to understanding the biology of SNO modifications is a reliable, sensitive and specific analytical method to measure SNO. This work addresses this challenge and advances in this area would be of broad interest.

The Cys-BOOST method is constructed from several of the key tools that have facilitated recent advances in the analysis of cysteine redox modifications, including Click chemistry probes, cleavable biotin linkers and latest generation mass spectrometry (MS) instruments. An important advantage is that the Click chemistry/capture is done at the peptide level, thereby ensuring an enriched population of Cys peptides and site localization by MS/MS. Despite these improvements, the method is still a biotin-switch approach, which has the advantages and disadvantages of this family of methodologies. The key step conferring specificity for SNO is the reductive removal of SNO with ascorbate, with rapid capture of the liberated thiol with an alkynyl iodoacetamide analog. A limitation of this approach is the possible interference by S-sulfenyl (SOH) sites, which could be reduced and labeled along with genuine SNO. Despite this concern, the analytical work appears to be competently done and the analytical platform presents no major concerns.

>>> We thank this reviewer for the positive comments on our work.

The first part of the work is a demonstration of the performance of the Cys-BOOST platform (minus the SNO capture) as a method to analyze cysteinyl peptides. The authors demonstrate that the Click capture methodology outperforms a thiol-labeling and immunoprecipitation method (Iodo-TMT). Good, but not really newsworthy.

>>> We agree that the reliance of anti-TMT enrichment on immunoprecipitation represents a significant drawback. Nevertheless, currently iodoTMT is still one of the most widely used strategies for Cys PTM analysis, probably as it is commercially available and relatively straightforward. For this reason we originally established the method in our lab, however, we were not pleased with its performance (and cost) which was the actual motivation to develop Cys-BOOST.

The major limitation of this work is that the authors simply treat a HeLa cell lysate with S-nitrosylglutathione (GSNO) and analyze the SNO peptides and sites. They generated a large dataset of putative SNO protein/peptide targets and sites and undertook motif and network analyses to interpret the possible significance of SNO sites. This all seems premature and of perhaps uncertain significance because the labeling experiment itself is of uncertain significance. By mixing HeLa lysate with 200 μ M GSNO, they get a lot of labeled sites, but it's not clear which of these might actually be SNO sites in an intact cell in a defined physiologic state. Indeed, the challenge of mapping and quantifying SNO in situ (i.e., in intact cells) is a more relevant, significant and certainly difficult challenge. It seems possible that the Cys-BOOST platform might be adapted to the in situ SNO mapping problem, but the authors haven't

attempted that yet. What the authors present here is a limited technical demonstration—a milestone on the way to a possibly significant achievement.

>>> We agree with reviewer #1 (and #2) that the original experiment lacked physiological significance. We therefore conducted a new experiment using SH-SY5Y cells. The neuroblastoma SH-SY5Y cell line is widely used as neuronal *in vitro* system, for instance in Parkinson's disease and Alzheimer's disease research. We treated SH-SY5Y cells with 100 μ M of S-Nitroso-N-Acetyl-D,L-Penicillamine (SNAP), a cell-permeable NO donor, for 5 min and compared to untreated controls. The neuron-like nature of SH-SY5Y cells allowed us to study the effect of SNO under neuropathological relevant conditions, i.e. early nitrosative stress. In this experiment, we could identify 2,158 unique *in-vivo* SNO sites on 1,443 proteins. From 1,859 confidently quantified SNO peptides, 5.6 % showed significant regulation after SNAP treatment, including SNO sites on proteins involved in neurogenesis, axonogenesis, glutamatergic synaptic transmission, cadherin binding, NADH metabolic process, protein folding, translation, and DNA replication. We added all relevant information on the experiment to the experimental procedures, results and discussion sections. Figure 6 shown below summarizes the results.

Fig. 6. SNO analysis of SNAP-treated and non-treated (control) SH-SY5Y cells. Top 20 enriched Gene Ontology (GO) terms of the SH-SY5Y SNO proteome: a. molecular function and b. biological process. c. Volcano plot of SNO peptides in SNAP-treated vs non-treated samples. Green dots ($n=39$) represent significantly downregulated (≥ 1.5 -fold, p -value ≤ 0.05) and blue dots ($n=65$) significantly upregulated (≥ 1.5 -fold, p -value ≤ 0.05) SNO peptides. d. High confidence STRING network of proteins with significantly changed (≥ 1.5 -fold, p -value ≤ 0.05) SNO levels. Proteins with upregulated SNO sites are marked blue, with downregulated green and with both up- and down-regulated red.

Minor point: Putative SNO peptides are apparently those identified with the linker fragment from the cleaved biotin tag. Although the mass shift would be distinctive, such fragments often yield diagnostic product ions in MS/MS (see ref. 20). Do the labeled peptides in the Cys-BOOST workflow yield such characteristic, tag-derived fragment ions? If so, can these be used to enhance the confidence of the assignments?

>>> We thank the reviewer for pointing this out. Indeed, the linker fragment from the cleaved biotin tag in Cys-BOOST leads to the diagnostic fragment ion (DFI) m/z 230.17 corresponding to the DFI-1 (m/z

368.16) of DYn-2-triazo-hexanoic-acid modified peptides. This DFI can be used to enhance the confidence of the assignments of HCD spectra of SNO peptides, as referred to in the revised manuscript:

Moreover, the linker fragment from the cleaved biotin tag in Cys-BOOST leads to the diagnostic fragment ion (m/z 230.17), which, if necessary, could be used to enhance the confidence of HCD spectra assignments in accordance with Yang et al. [25].

Reviewer #2:

In their manuscript, Mnatsakanya et al. describe a modified switch technique for the detection of protein S-nitrosothiols (SNOs). Thousands of S-nitrosylated proteins are identified. Inasmuch as cells express ~10,000 proteins, the data imply that the bulk of the proteome may be subject to SNO control. This point may ultimately merit publication, but the data can be strengthened through additional experimentation.

>>> We thank the reviewer for thorough assessment and the positive opinion on our manuscript.

Major Points

The strategy follows the well-established switch principle for the detection of SNO-proteins. However, rather than directly labeling with thiol-reactive biotin, the authors “switch” with IAA-alkyne, which ultimately reacts with a hydrazine-cleavable biotin linker. The method allows for off-bead protein digestion prior to labeling. The authors also validate differential GSNO reactivity motifs, namely acid-base and hydrophobic motifs. Importantly, the authors claim increased sensitivity compared with other MS-based techniques for SNO-protein identification. In support of this view, they identify and collate by far the largest set of S-nitrosylated proteins to date. However, three experiments are needed to support the claims for enhanced sensitivity and physiological relevance: 1. SNO-Cys enrichment and identification should be compared with another technique (e.g. biotin switch or SNO-RAC); 2. an Ascorbate (–) control should be provided; and 3. the authors should use the method to detect endogenous SNO-proteins (e.g. SNO generated in RAW cells stimulated with LPS/IFN or HEK cells transfected with nNOS or eNOS.) Knockdown of GSNO reductase (or tissues from GSNOR mice) would identify SNO-proteins formed by GSNO.

Specifically:

1. The authors should directly compare the Cys-BOOST method with iodoTMT or biotin switch (or SNO-RAC) methods for SNO-Cys identification in order to demonstrate direct superiority of this method.

>>> From our experience with the enrichment of other PTMs using different protocols (e.g. phosphorylation with TiO₂, SCX, IMAC, ERLIC, different pTyr antibodies, SH2 domains; glycosylation with TiO₂, ENSAS/SCX, lectin affinity chromatography; ubiquitination with antibodies; proteolytic cleavage/N-terminomics with COFRADIC, ChaFRADIC; AMPylation with click chemistry and Lys-acetylation with different antibodies; as well as the enrichment of cross-linked peptides and enrichment using specific anti-peptide antibodies, we learned that these protocols are often not readily established in a way that allows a fair and systematic comparison, such as we have presented here for Cys-BOOST and iodoTMT.

Indeed, we first established iodoTMT in our lab to study Cys modifications. Even after extensive optimization of the manufacturer's protocol (that worked poorly in our hands) leading to considerable improvement of iodoTMT's performance, we were not pleased with the method, due to various reasons such as recovery, specificity but also cost. We therefore developed Cys-BOOST and decided to present a fair comparison with a well-performing iodoTMT protocol.

Based on our experience with enrichment protocols we believe that it would have not been possible to conduct a fair comparison with a completely novel method that we had to establish from scratch. A method, for which we would have to purchase all the required consumables, establish the protocol, test its performance in order to conduct a well-designed and fair comparison. Moreover, SNO-RAC requires on-bead digestion prior to isobaric tagging, a step that itself would need extensive optimization and validation for a fair comparison. We therefore sincerely apologize that we could not find a way how to address this reviewer's comment in a reasonable and timely manner.

2. In standard SNO switch methods, -ascorbate is used as a control for blocking efficiency. However the authors do not control for blocking efficiency in SNO-Cys enrichment, but only perform MS on +ascorbate samples, raising the concern that many of the proteins and sites they identify are due to incomplete block of free Cys. The -ascorbate control also serves to control for proteins/peptides that non-specifically bind to streptavidin beads. The authors should repeat SNO-Cys MS experiments comparing +/- ascorbate in order to demonstrate that their reported hit sensitivity is not due to incomplete blocking of free Cys thiols and non-specific bead binding.

>>> We agree with this reviewer and apologize for the absence of this important control experiment in the original manuscript. For the revised manuscript, we have conducted 3 independent experiments that enable us to address the issue: (i) -ascorbate as a control for GSNO-treated HeLa lysate, (ii) applying increasing concentrations (5 mM, 25 mM, 50 mM and 100 mM) of the blocking reagent for labeling of total free Cys in HeLa cells, followed by fluorescence detection of the saturation of the labeling, (iii) analysis of the completeness of free Cys blocking by IAA in SH-SY5Y cell lysates. All 3 experiments confirmed the high efficiency of free Cys blocking with 100 mM IAA and are summarized in the new supplemental figures 3-5.

We added the sentence: *"To exclude the presence of free Cys prior to ascorbate reduction, leading to false positive identification of SNO sites, we verified the completeness of free Cys blocking in the initial step of our ST conditions (Sup. Fig.3-5)."* to the results section of the revised manuscript. Supplemental figures 3-5 are presented below.

Notably, the issue of false positive identifications as a result of a non-specific binding is not relevant for Cys-BOOST, as the enrichment is not based on the disulfide-exchange and post-elution Cys alkylation (e.g. SNO-RAC), but on the chemical cleavage of the bioorthogonal linker, thus all enriched peptides contain the linker fragment.

Sup. Fig. 3. Analysis of SNO reduction conditions (20 mM Na ascorbate (+asc), 10 min UV (UV) and control (– Na ascorbate (–asc)) in 200 μ M GSNO-treated HeLa lysates. SNO peptides were analyzed according to the Cys-BOOST workflow. Free Cys were blocked with 100 mM IAA, excess of the reagent was removed by double acetone precipitation. SNO was switched with 3 mM IAA-alkyne (IAA-alk) in the presence (+asc) or absence (–asc) of 20 mM Na ascorbate or after 10 min exposure to UV light. After tryptic digestion, TMT labeling, enrichment and on-tip pH 10 fractionation, the SNO peptides were analyzed by LC-MS3 in synchronous precursor selection (SPS) mode. The data was analyzed using Proteome discoverer (PD) v2.2. No normalization of total protein amount was applied, as low (< 5 %) RSDs of reporter ion intensities

observed for background peptides present in the Cys-BOOST flow through (1 μ g measured by LC-SPS), showed that the protein amounts in all samples were equivalent (Sup. Table 5a). The amount of switched SNO peptides was relatively quantified in each sample based on the TMT reporter ion intensities (S/N). Boxplots of S/N of 7575 SNO peptide spectra match (PSMs) considered for quantification (average reporter S/N threshold ≥ 10 , without carbamidomethylation (CAM) on Cys and oxidation on Met) show comparable S/N for +asc and UV reduction. The mean (\square) S/N of the –asc samples (n=3) is 2.6, the median is 0.56, with an average of 7.2 % of PSMs having S/N ≥ 10 (Sup. Table 5b). Accordingly, the –asc control indicates the near completeness of the first blocking step using 100 mM IAA. The weak remaining signals in –asc samples, mostly below the limit of quantification (LOQ S/N ≥ 10), can be most likely attributed to the SNO reduction by negligible exposure to light.

Sup. Fig. 4. Total free Cys labeling with increasing concentrations of IAA-alk. Total Cys of HeLa cell lysates (2 mg/mL) were reduced with 5 mM TCEP for 1h at 56 $^{\circ}$ C, the pH of the samples was adjusted with 200 mM HEPES (pH 7.5), then the samples were labeled with 5 mM, 25 mM, 50 mM or 100 mM IAA-alk for 30 min at 25 $^{\circ}$ C. Excess of the reagents was removed by acetone precipitation and fluorescent 5-TAMRA-azide (Carl Roth) was introduced using CuAAC. 5 μ g of each lysate was separated by SDS-PAGE, followed by fluorescence detection at 580 nm (Typhoon Trio, GE Healthcare). Afterwards, the gel was stained with Coomassie Brilliant Blue solution (the gel not shown). Fluorescence signal intensities and protein abundance (based on Coomassie staining) were quantified using ImageJ. The fluorescence signal shows on average a 4.6 % increase in 100 mM compared to 50 mM IAA-alk labeled samples (b), indicating the saturation of labeling at 100 mM. The fluorescence signal was normalized to protein abundance (Coomassie staining) in each lane, error bars show the SD of n=3 replicates.

Sup. Fig. 5. Analysis of the completeness of IAA blocking of free Cys in the initial step of the ST. The free Cys in SH-SY5Y cell lysates (2 mg/mL in HENS) were blocked with 25 mM, 50 mM or 100 mM IAA for 30 min at 25 °C. Excess of the reagents was removed by double acetone precipitation. The pellets were resolubilized and a second labeling step with 10 mM IAA-alk was applied for blocking of the remaining free Cys. After tryptic digestion and TMT labeling the IAA-alk labeled peptides were enriched using our Cys-BOOST workflow. The eluate was analyzed by LC-SPS. The data was analyzed using Proteome discoverer (PD) v2.2. No normalization of total protein amount was applied, as low (< 5 %) RSD of reporter ion intensities observed for background peptides present in the Cys-BOOST flow through (1 µg

measured by LC-SPS), shows that the protein amounts in all samples were equivalent (Sup. Table 6a). The amount of the remaining free Cys after IAA blocking was relatively quantified in each sample based on the TMT reporter ion intensities (S/N). Boxplots of S/N of 2521 PSMs considered for quantification (average reporter S/N threshold ≥ 10 , without CAM on Cys and oxidation on Met) show drastic reduction of remaining free Cys after 50 mM and 100 mM IAA blocking compared to 25 mM. The mean (\square) S/N is 6.3 and 0.4, the median is 0.9 and 0, with on average 15 % and 1% of PSMs having S/N ≥ 10 in 50 mM and 100mM IAA blocked samples respectively (Sup. Table 6b). Hence, confirming the completeness of the free Cys blocking with 100 mM IAA.

3. Exogenous GSNO is representative of nitrosative stress. A focus on endogenously regulated SNO-proteins would provide a more physiological picture; knockdown of GSNO reductase would identify endogenous GSNO-derived SNO-proteins. The authors should compare SNO-Cys from a system in which endogenous SNO levels would be altered (e.g. WT and GSNOR-KO tissue stimulated with LPS/IFN) and, if necessary, knock down GSNOR to assess the role of endogenous GSNO.

>>> We agree with Reviewer #2 (and #1) that the original experiment lacked physiological significance. We therefore conducted a new experiment using SH-SY5Y cells. The neuroblastoma SH-SY5Y cell line is widely used as neuronal *in vitro* system, for instance in Parkinson's disease and Alzheimer's disease research. We treated SH-SY5Y cells with 100 µM of S-Nitroso-N-Acetyl-D,L-Penicillamine (SNAP), a cell-permeable NO donor, for 5 min and compared to untreated controls. The neuron-like nature of SH-SY5Y cells allowed us to study the effect of SNO under neuropathological relevant conditions, i.e. early nitrosative stress. In this experiment, we could identify 2,158 unique *in-vivo* SNO sites on 1,443 proteins. From 1,859 confidently quantified SNO peptides, 5.6 % of showed significant regulation after SNAP treatment, including SNO sites on proteins involved in neurogenesis, axonogenesis, glutamatergic synaptic transmission, cadherin binding, NADH metabolic process, protein folding, translation, and DNA replication. We added all relevant information on the experiment to the experimental procedures, results and discussion sections. Figure 6 shown below summarizes the results.

Fig. 6. SNO analysis of SNAP-treated and non-treated (control) SH-SY5Y cells. Top 20 enriched Gene Ontology (GO) terms of the SH-SY5Y SNO proteome: *a.* molecular function and *b.* biological process. *c.* Volcano plot of SNO peptides in SNAP-treated vs non-treated samples. Green dots ($n=39$) represent significantly downregulated (≥ 1.5 -fold, p -value ≤ 0.05) and blue dots ($n=65$) significantly upregulated (≥ 1.5 -fold, p -value ≤ 0.05) SNO peptides. *d.* High confidence STRING network of proteins with significantly changed (≥ 1.5 -fold, p -value ≤ 0.05) SNO levels. Proteins with upregulated SNO sites are marked blue, with downregulated green and with both up- and down-regulated red.

4. The authors provide a strategy for assessing potential bias of mass-spec identification, by comparing their nitrosylated protein dataset to a HeLa cell proteomic database with quantified levels of protein abundance. However, they compare only the top 500 proteins in each list to determine that their technique identifies proteins over a 5-order of magnitude difference in expression level. The authors should also compare the complete datasets in order to derive a complete assessment of just how wide a range of proteins is identified by their technique.

>>> We fully agree with the reviewer that a full comparison is more meaningful. We therefore compared all 6120 proteins quantified in our HeLa proteome dataset (at least one unique peptides, 1% protein false discovery rate) with the HeLa SNO proteome, as before based on NSAF values. Our data clearly demonstrate that we cover the “whole” proteome and there is no significant bias towards high abundance proteins. Indeed, the SNO proteome data comprise more than 600 proteins that were not quantified based on unique peptides in the global proteome, indicating low abundance for the majority of these. We added corresponding figures to the supplement of the revised manuscript (suppl. figure 1) with the following figure caption.

Sup. Fig. 1. Good coverage of the a. HeLa and b. SH-SY5Y SNO proteome. Proteome dynamic range represented by NSAF values (blue): Only proteins with at least 1 unique peptide (1% protein false discovery rate) were considered and ordered in descending abundance, as represented by descending NSAF values, corresponding to 6120 (HeLa) and 6294 (SH-SY5Y) proteins, respectively. SNO proteome (red): The NSAF values for proteins with quantified high confidence SNO sites are shown as red dots, corresponding to 3007 SNO proteins for HeLa and 1413 SNO proteins for SH-SY5Y. Notably, an additional 625 (HeLa) and 30 (SH-SY5Y) proteins with SNO sites are not represented in the graphs, as they were not detected with unique peptides in the global proteomes and therefore lack an NSAF value, indicating a presumably

low abundance of this proteins. The quantified SNO sites clearly cover the whole dynamic range of the corresponding proteomes. Top: the number of quantified SNO sites per individual protein is plotted, representing 6247 HeLa and 2158 SH-SY5Y SNO sites.

We changed the corresponding section in the results section of the main text accordingly:

Mapping the identified SNO proteins to the HeLa proteome based on the normalized abundance factors (NSAF) [40], showed that the SNO proteome covers the whole dynamic range of the HeLa proteome (Sup. Fig. 1a).

5. The authors frame the issue of protein S-nitrosylation as the result of largely non-enzymatic processes, in both the introduction and discussion, ignoring a large and growing literature on enzymes that convert NO to SNO, those that transfer SNO between proteins, and those that remove SNO from proteins and small molecules. Indeed, confusingly, the authors mention some of this literature in the discussion, but frame it in an unusual way. The authors need to reassess the current literature of SNO formation, and update their introduction and discussion accordingly.

>>> We thank the reviewer for this important remark. We updated our manuscript according to the most recent literature and added this section to the introduction of the revised manuscript.

The enzymatic nature of SNO and denitrosylation is increasingly appreciated in target selectivity and specificity of SNO-mediated redox signaling [14, 15]. A growing number of SNO proteins are recognized as transnitrosylases (NO donors) that transfer their NO group to the free Cys of target proteins (NO acceptors) [16-18]. SNO-free thiol homeostasis is maintained by the protein denitrosylase thioredoxin (Trx) system and specialized low-molecular-weight SNO (i.e. S-nitrosoglutathione (GSNO) and SNO-coenzyme) reductase systems [14].

6. The authors assume the GSNO mediates S-nitrosylation by transnitrosylation. But they provide no data to support this. Also GSNO will not cross the cell directly. Remove (trans) from title.

>>> We agree with this reviewer that the term transnitrosylation rather represents protein-to-protein SNO transfer, as it involves transnitrosylase enzymatic activity. Although in the early papers of SNO research pioneers J. S. Stamler and S. A. Lipton thiol-to thiol transfer is often referred as transnitrosylation. Accordingly, we removed the “trans” from the title and in the main text which now reads

Proteome-wide detection of cysteine nitrosylation targets and motifs using bioorthogonal cleavable-linker-based enrichment and switch technique (Cys-BOOST)

Reviewer #3 (Remarks to the Author):

This manuscript describes a method to enrich peptides containing free cysteines. The tool is aimed at deciphering redox chemistry which is a very important for regulation.

The authors are modifying the free cysteines with an alkyne to which, later in the process, a base-labile biotin unit is attached which allows for ‘catch-and-release’.

The report is divided into two parts where the first compares this new strategy to a commercial product (IodoTMT) and in the 2nd part the authors are applying the strategy to a biological question. Both experiments are well executed.

>>> We thank this reviewer for the positive opinion on our manuscript.

Benchmarking: The authors used a fully reduced HeLa lysate to test enrichment. To address technical variability the authors used triplicate. The presented method yielded 25,000 cysteine containing peptides with an enrichment of 98% vs the iodo-TMT strategy that resulted in “only” 10,000 peptides with a much lower enrichment (70%). The authors are naming sensitivity as a reason for the better performance. A Venn diagram in Fig. 2 for the compared approaches will be useful.

>>> We added the Venn diagram showing the overlap of the 25,019 Cys peptides enriched by Cys-BOOST with the 9,966 Cys peptides enriched by iodoTMT as supplemental figure 7 (below) and refer to this in the revised results section:

Additionally, the total Cys peptides enriched by Cys-BOOST showed 69% overlap with iodoTMT peptides (Sup. Fig. 7).

Sup. Fig. 7. Venn diagram showing the overlap of total Cys peptides enriched by iodoTMT and Cys-BOOST.

Though the antibody based iodo-TMT also is an approach to measure free cysteines many antibody based techniques requires larger starting amounts and are generally less sensitivity. ICAT has been around for many years and I find that a better benchmarking of the proposed strategy would be to compare to such an experiments. Using literature – if available – would be sufficient.

>>> Based on this reviewer’s suggestion we added a table as supplemental figure 6 that summarizes SNO studies conducted using different methods in comparable settings.

Method	Model system	Number of SNO proteins/peptides/sites	Amount of starting material/condition [mg]	Quantification	Year	Reference
RAC	CysNO treated HEK293 cells	398 peptides	1	iTRAQ; MS2	2009	[2]
SNO-RAC	GSNO-treated mouse heart homogenates	951 proteins, ~2000 sites	1	Label-free; MS1	2011	[3]
SNO-RAC	GSNO-treated mouse skeletal muscle homogenates	488 sites	0.5	iTRAQ; MS2	2013	[4]
SNO-RAC	Cys-NO treated nuclear extracts of rat cortical neurons	614 proteins	0.4	N/a	2018	[5]
SNO ICAT	SNO-Trx1-treated SH-SY5Y cell lysate	50-76 sites	0.3	Light and heavy ICAT; MS1	2011	[6]
SNO ICAT	normoxic mouse heart	907 sites	N/a	Light and heavy ICAT; MS1	2017	[7]
iodoTMT	CysNO-treated BV-2 cells; LPS-stimulated BV-2 cells	134 sites; 101 sites	0.4	TMT; MS2	2014	[8]
iodoTMT	rat cardiomyocyte under hypoxia	169 proteins, 266 sites	0.3	TMT; MS2	2014	[9]
HPDP-biotin	CysNO-treated NPrEC cells	81 sites	1	N/a	2010	[10]
HPDP-biotin	LPS and IFN- γ -treated RAW264.7 cells	156 proteins	1	SILAC; MS1	2012	[11]
CysPAT	GSNO treated RAW 264.7 cell extracts	795 proteins, 1450 peptides	0.4	N/a	2018	[12]
Cys-BOOST	GSNO-treated HeLa cell extracts	3632 proteins, 8304 sites	0.3	TMT; MS2	2019	current study
Cys-BOOST	SNAP-treated SH-SY5Y cells	1443 proteins, 2158 sites	0.25	TMT; MS3	2019	current study

Sup. Fig. 6. Table of comparison of SNO studies.

My only concern re. the experiment is how efficient the iodo-alkyne reagent is removed. Such compounds are ultra-reactive and if not removed completely they can result in unwanted alkylation of reduced di-sulfur bridges upon addition of TCEP leading to unwanted side effects and false positives. Can TCEP and IAA (IAA would be in excess of iodo-alkyne) be added simultaneously to minimize the above? I do appreciate that two 'removal steps' are included but including some considerations would be appropriate when discussing the strategy.

>>> We agree with this reviewer that the iodoacetyl compounds are highly reactive and even trace amounts can result in false positives. In Cys-BOOST this issue is handled by dissolving the protein pellet, after IAA-alkyne removal, in 1 % aqueous SDS. In this condition addition of 5 mM TCEP reduces the pH of the solution to 2-3, which is the most optimal pH for TCEP-mediated Cys reduction. It is known that at pH 2-3 the formation of thiolate anion (necessary for Cys alkylation) is highly unlikely and even in the

presence of huge excess of iodoacetyl reagent no quantitative reaction can be achieved. After the reduction, the pH of the reaction is adjusted to pH 7.5 using 200 mM HEPES. Only then the alkylation with 12 mM IAA is possible. We emphasized the pH adjustment step with 200 mM HEPES to pH 7.5 in the revised version of our manuscript.

The pellets were dissolved in 100 μL of 1 % SDS. The following steps of reduction with 5 mM TCEP, adjustment of pH with 200 mM HEPES pH 7.5, alkylation with 12 mM IAA, tryptic digestion, TMT 10plex™ labeling and desalting were executed as described above for total Cys analysis by Cys-BOOST.

Maybe just a display of my ignorance but how does GSNO work? This is an important part of this experiment.

>>> We apologize for not providing an explanation of the GSNO-mediated SNO formation in the original manuscript. We now added this important information to the introduction of our revised manuscript.

GSNO participates in protein SNO formation mainly through thiol-to-thiol NO⁺ transfer. For the bacterial transcription factor OxyR, GSNO-mediated nitrosylation was shown to be facilitated by a sequence motif including flanking acidic/basic and hydrophobic residues [13].

If space is a concern, the large section describing the biological findings, using STRING (two full figures) can seem ‘light’ compared to some of the Nature Comm. reports I have looked at.

>>> Based on this reviewer’s suggestion we combined the two STRING figures into a single one (new figure 4 below).

Fig. 4. The STRING networks of proteins with a. significantly downregulated, b. the 500 strongest upregulated SNO sites indicate the association of SNO to splicing, translation, ubiquitin mediated proteolysis, DNA replication, cellular response to stress, vesicle mediated transport, GTPase mediated signaling and aminoacyl-tRNA biosynthesis pathways.

REVIEWERS' COMMENTS:

Reviewer #1 (Remarks to the Author):

The authors have made substantive revisions in response to all of my comments and most of those of the other reviewers. Most important from my perspective is their demonstration that the method can quantify SNO protein modifications in intact cells. This manuscript will be a useful contribution to our expanding analytical repertoire for cysteine redox studies in biological systems.

Reviewer #2 (Remarks to the Author):

The authors have addressed adequately most points and the paper has been improved. The paper includes the most comprehensive and expansive list of SNO-proteins in the literature including new insights. The findings suggest that the majority of the proteome is likely subject to this PTM (note: active denitrosylation determines steady state in same way that phosphatases determine steady state for phosphorylation). The abstract can be significantly improved (see below).

Point 3. The authors have misunderstood the original concern. The question was, if this new Cys-Boost method is indeed more sensitive than other techniques (particularly iodoTMT, since the text refers to it specifically), it should be useful for identifying SNO sites under conditions of normal NO production. Importantly, the authors have now performed assays in two cell types showing a surprisingly large numbers of SNO-proteins in BASAL conditions. The authors should explicitly include the results under basal conditions in the abstract and compare the basal levels for their experiments in the Discussion, highlighting the large number of sites identified in the absence of any exogenous NO source. It should be noted that diverse stimuli will increase selectively SNO levels and the basal levels are regulated by denitrosylases.

Point 4. The authors have extended their comparison of SNO-protein detection versus protein abundance in the cell, and now provide compelling evidence that their technique has high sensitivity to detect low-abundance proteins. Bias against low abundance proteins is evident particularly for the SH-SY5Y cell samples (and to a lesser extent in HeLa cells), as the density of SNO-sites is reduced as proteins become more rare, but it is also obvious that SNO sites are still identified among even the rarest low-abundance proteins (Supp Fig 1). Since the report is much more about method development than in breaking new conceptual ground, this informative quantitative comparison should be highlighted in the main text rather than relegated to the supplementary information. It

would be very informative to assess depth of coverage vs protein abundance for just the SNO sites identified under basal conditions.

Additional points.

1. The term S-nitrosylation should be used in title or abstract at least once (because this is more commonly used than cysteine nitrosylation and will improve access).

2. See GSNO motif in "Host S-nitrosylation inhibits clostridial small molecule-activated glucosylating toxins. Savidge T et al Nat Med. 2011 Aug 21;17(9):1136-41. doi: 10.1038/nm.2405.

3. See hmp work in Mol Cell (Seth et al) on novel SNO machinery

The abstract can be significantly improved. Consider the following changes.

S-nitrosylation has emerged as a major player in cellular signaling but remains challenging to assay. Here, we present Cys-BOOST, a novel chemical proteomics strategy for quantitative analysis of Cys nitrosylation sites, identifying thousands of sites across a large part of the proteome. Cys-BOOST-based analysis of Cys nitrosylation (SNO) in S-nitrosoglutathione (GSNO)-treated and non-treated HeLa cells (i.e basal conditions) allowed the identification of unprecedented numbers of SNO proteins (3,632), peptides (9,314) and sites (8,304), including X sites under basal conditions, covering a wide dynamic range of the HeLa proteome. Consensus motifs of SNO sites with differential reactivity to GSNO confirmed the relevance of both acid-base catalysis and local hydrophobicity for NO-targeting to particular Cys. In-vivo SNO analysis of SH-SY5Y cells revealed proteins involved in neuro(axono)genesis, glutamatergic synaptic transmission, cadherin binding, NADH metabolic process, protein folding, translation, and DNA replication, suggesting coverage in many if not most cellular functions. Our work suggests that S-nitrosylation is a global regulator of protein function akin to phosphorylation and ubiquitylation and provides a new tool to enable deep analysis.

Reviewer #3 (Remarks to the Author):

I have read the rebuttal and re-read most of the manuscript. I believe that the authors has addressed my concerns and I am happy to accept as is.

REVIEWERS' COMMENTS:

Reviewer #1 (Remarks to the Author):

The authors have made substantive revisions in response to all of my comments and most of those of the other reviewers. Most important from my perspective is their demonstration that the method can quantify SNO protein modifications in intact cells. This manuscript will be a useful contribution to our expanding analytical repertoire for cysteine redox studies in biological systems.

>>>> We thank reviewer #1 for the positive comments on our work!

Reviewer #2 (Remarks to the Author):

The authors have addressed adequately most points and the paper has been improved. The paper includes the most comprehensive and expansive list of SNO-proteins in the literature including new insights. The findings suggest that the majority of the proteome is likely subject to this PTM (note: active denitrosylation determines steady state in same way that phosphatases determine steady state for phosphorylation). The abstract can be significantly improved (see below).

>>>> We thank reviewer #2 for the positive comments on our work!

Point 3. The authors have misunderstood the original concern. The question was, if this new Cys-Boost method is indeed more sensitive than other techniques (particularly iodoTMT, since the text refers to it specifically), it should be useful for identifying SNO sites under conditions of normal NO production. Importantly, the authors have now performed assays in two cell types showing a surprisingly large numbers of SNO-proteins in BASAL conditions. The authors should explicitly include the results under basal conditions in the abstract and compare the basal levels for their experiments in the Discussion, highlighting the large number of sites identified in the absence of any exogenous NO source. It should be noted that diverse stimuli will increase selectively SNO levels and the basal levels are regulated by denitrosylases.

>>>> We thank the reviewer for pointing this out. In agreement with the reviewer's suggestions to improve the abstract we now mention the detection of basal SNO and also added a short section in the discussion (see also point 4 below).

Point 4. The authors have extended their comparison of SNO-protein detection versus protein abundance in the cell, and now provide compelling evidence that their technique has high sensitivity to detect low-abundance proteins. Bias against low abundance proteins is evident particularly for the SH-SY5Y cell samples (and to a lesser extent in HeLa cells), as the density of SNO-sites is reduced as proteins become more rare, but it is also obvious that SNO sites are still identified among even the rarest low-abundance proteins (Supp Fig 1). Since the report is much more about method development than in breaking new conceptual ground, this informative quantitative comparison should be highlighted in the main text rather than relegated to the supplementary information. It would be very informative to assess depth of coverage vs protein abundance for just the SNO sites identified under basal conditions.

>>>> We stress this finding now more in the revised version.

In the Results:

2,151 SNO sites were quantified in both, control (basal) and SNAP-treated samples, covering four orders of magnitude of the dynamic range of the SH-SY5Y proteome (Supplementary Data 4c, Supplementary Figure 51b).

In the Discussion:

As exemplified in SH-SY5Y cells, Cys-BOOST allows the proteome-wide identification of SNO even from low abundance proteins and under basal conditions.

...and in the abstract:

Applying Cys-BOOST to SH-SY5Y cells we identify 2,151 SNO sites under basal conditions and reveal significantly changed SNO levels as response to early nitrosative stress, involving neuro(axono)genesis, glutamatergic synaptic transmission, protein folding/translation, and DNA replication. Our work suggests SNO as a global regulator of protein function akin to phosphorylation and ubiquitination.

Additional points.

1. The term S-nitrosylation should be used in title or abstract at least once (because this is more commonly used than cysteine nitrosylation and will improve access). 2. See GSNO motif in "Host S-nitrosylation inhibits clostridial small molecule-activated glucosylating toxins. Savidge T et al Nat Med. 2011 Aug 21;17(9):1136-41. doi: 10.1038/nm.2405. 3. See hmp work in Mol Cell (Seth et al) on novel SNO machinery

>>>> We replaced the term Cys(teine) nitrosylation with S-nitrosylation.

The abstract can be significantly improved. Consider the following changes.

S-nitrosylation has emerged as a major player in cellular signaling but remains challenging to assay. Here, we present Cys-BOOST, a novel chemical proteomics strategy for quantitative analysis of Cys nitrosylation sites, identifying thousands of sites across a large part of the proteome. Cys-BOOST-based analysis of Cys nitrosylation (SNO) in S-nitrosoglutathione (GSNO)-treated and non-treated HeLa cells (i.e. basal conditions) allowed the identification of unprecedented numbers of SNO proteins (3,632), peptides (9,314) and sites (8,304), including X sites under basal conditions, covering a wide dynamic range of the HeLa proteome. Consensus motifs of SNO sites with differential reactivity to GSNO confirmed the relevance of both acid-base catalysis and local hydrophobicity for NO-targeting to particular Cys. In-vivo SNO analysis of SH-SY5Y cells revealed proteins involved in neuro(axono)genesis, glutamatergic synaptic transmission, cadherin binding, NADH metabolic process, protein folding, translation, and DNA replication, suggesting coverage in many if not most cellular functions. Our work suggests that S-nitrosylation is a global regulator of protein function akin to phosphorylation and ubiquitylation and provides a new tool to enable deep analysis.

>>>> We have improved our abstracts in accordance with reviewer #2's suggestion, but had to meet the word limit of 150. We also would like to point out that we did not want to limit the abstract solely to S-nitrosylation, as we want to stress here, that Cys-BOOST allows the analysis of other reversible Cys modifications as well, depending on the applied switch technique.

The abstract (exactly 150 words) now reads:

Cysteine modifications emerge as important players in cellular signaling and homeostasis. Here, we present a chemical proteomics strategy for quantitative analysis of reversibly modified Cysteines using bioorthogonal-cleavable-linker and switch technique (Cys-BOOST). Compared to iodo-TMT for total Cysteine analysis Cys-BOOST shows a threefold higher sensitivity and considerably higher specificity and precision. Analyzing S-nitrosylation (SNO) in S-nitrosoglutathione (GSNO)-treated and non-treated HeLa extracts Cys-BOOST identifies 8,304 SNO sites on 3,632 proteins covering a wide dynamic range of the proteome. Consensus motifs of SNO sites with differential GSNO-reactivity confirm the relevance of both acid-base catalysis and local hydrophobicity for NO-targeting to particular Cysteines. Applying Cys-BOOST to SH-SY5Y cells we identify 2,151 SNO sites under basal conditions and reveal significantly changed SNO levels as response to early nitrosative stress, involving neuro(axono)genesis, glutamatergic synaptic transmission, protein folding/translation, and DNA replication. Our work suggests SNO as a global regulator of protein function akin to phosphorylation and ubiquitination.

Reviewer #3 (Remarks to the Author):

I have read the rebuttal and re-read most of the manuscript. I believe that the authors has addressed my concerns and I am happy to accept as is.

>>>> We thank reviewer #3 for the positive feedback!